

# Balloon-borne measurements of temperature, water vapor, ozone and aerosol backscatter at the southern slopes of the Himalayas during StratoClim 2016-2017

Simone Brunamonti[1], Teresa Jorge[1], Peter Oelsner[2], Sreeharsha Hanumanthu[3,4], Bhupendra B. Singh[3], K. Ravi Kumar[3,9], Sunil Sonbawne[3], Susanne Meier[2], Deepak Singh[5], Frank G. Wienhold[1], Bei Ping Luo[1], Maxi Böttcher[1], Yann Poltera[1], Hannu Jauhiainen[8], Rijan Kayastha[6], Ruud Dirksen[2], Manish Naja[5], Markus Rex[7], Suvarna Fadnavis[3] and Thomas Peter[1]

[1]Institute for Atmospheric and Climate Science (IAC), Swiss Federal Institute of Technology (ETH), Zürich, Switzerland
[2]Deutscher Wetterdienst (DWD) / GCOS Reference Upper Air Network (GRUAN) Lead Center, Lindenberg, Germany
[3]Indian Institute of Tropical Meteorology (IITM), Pune, India
[4]Forschungszentrum Jülich (FZJ), Jülich, Germany
[5]Aryabhatta Research Institute of Observational Sciences (ARIES), Nainital, India
[6]Kathmandu University (KU), Dhulikhel, Nepal
[7]Alfred Wegener Institute (AWI) for Polar and Marine Research, Potsdam, Germany
[8]Vaisala Oyj, Vantaa, Finland
[9]*Now at* King Abdullah University of Science and Technology (KAUST), Thuwal, Saudi Arabia

*Correspondence to:* Simone Brunamonti (simone.brunamonti@env.ethz.ch)

**Abstract.** The Asian summer monsoon anticyclone (ASMA) is a major meteorological system of the upper troposphere-lower stratosphere (UTLS) during boreal summer. It is known to be enriched in tropospheric trace gases and aerosols, due to rapid lifting from the boundary layer by deep convection and subsequent horizontal confinement. Given its dynamical structure, the ASMA offers a very efficient pathway for the transport of pollutants to the global stratosphere. Detailed understanding of the ASMA structure and processes requires accurate in-situ measurements. Here we present balloon-borne measurements of temperature, water vapor, ozone and aerosol backscatter conducted within the StratoClim project from two stations at the southern slopes of the Himalayas. In total we performed 63 balloon soundings during two main monsoon-season campaigns, in August 2016 in Nainital, India (NT16$_{AUG}$) and July-August 2017 in Dhulikhel, Nepal (DK17), and one brief post-monsoon campaign in Nainital in November 2016 (NT16$_{NOV}$). These measurements provide unprecedented insights into the ASMA thermal structure and its relations to the vertical distributions of water vapor, ozone and aerosols. To study the structure of the UTLS during the monsoon season, we adopt the thermal definition of tropical tropopause layer (TTL), and define the region of altitudes between the lapse rate minimum (LRM) and the cold-point tropopause (CPT) as the Asian Tropopause Transition Layer (ATTL). Further, based on air mass trajectories, we define the Top of Confinement (TOC) level of ASMA, which divides the lower stratosphere (LS) into a Confined LS (CLS), below the TOC and above the CPT, and a Free LS (FLS), above the TOC. Using these thermodynamically-significant boundaries, our analysis reveals that the composition of the UTLS is affected by deep convection up to altitudes 1.5-2 km above the CPT due to the horizontal confinement effect of ASMA. This is shown by





enhanced water vapor mixing ratios in the Confined LS compared to background stratospheric values in the Free LS, observed in both NT16$_{AUG}$ (+0.5 ppmv) and DK17 (+0.75 ppmv), and by enhanced aerosol backscatter of the Asian tropopause aerosol layer (ATAL) extending into the Confined LS, as observed in NT16$_{AUG}$. The CPT was 600 m higher in altitude and 5 K colder in DK17 compared to NT16$_{AUG}$ and strong ozone depletion was found in the ATTL and CLS in DK17, suggesting stronger

convective activity during DK17 compared to NT16$_{AUG}$. An isolated water vapor maximum in the Confined LS, about 1 km above the CPT, was also found in DK17, which we argue is due to overshooting convection hydrating the CLS. These evidence show that the vertical distributions and variability of water vapor, ozone and aerosols in the Asian UTLS are controlled by the top height of the anticyclonic confinement in ASMA, rather than by CPT height as in the conventional understanding of TTL, and suggest that the ASMA contributes to moistening the global stratosphere and to increase its aerosol burden.

**1. Introduction**

Large-scale deep convection associated with the Asian summer monsoon (ASM) during boreal summer induces a strong and persistent anticyclonic vortex in the upper troposphere - lower stratosphere (UTLS), known as ASM anticyclone (ASMA) (e.g. Park et al., 2007) or, previously, as Tibetan high (e.g. Krishnamurti and Bhalme, 1976). The ASMA is confined by the sub-tropical westerly jet stream to the north (40-45°N) and the equatorial easterly jet stream to the south (10-15°N), and spans

roughly one third of the northern hemisphere's longitudes (20-140°E), between the eastern Mediterranean Sea and the western Pacific Ocean. Its geographic center is above the Tibetan plateau and the altitude of maximum strength of the anticyclonic circulation is around the local tropopause (17-18 km), which is the highest worldwide during the ASM season (Dethof et al., 1999; Bian et al., 2012; Garny and Randel, 2016; Ploeger et al., 2015; Pan et al., 2016). The ASMA is subject to strong dynamical variability, influencing its areal extent, location and strength, oscillations and eddy shedding (Randel and Park,

2006; Yan et al., 2012; Garny and Randel, 2013; Vogel et al., 2014; Nützel et al., 2016).

From satellite measurements, the ASMA is known to be enriched in tropospheric trace species and pollutants, including water vapor, carbon monoxide, methane, hydrogen cyanide, peroxyacetil nitrate (Randel et al., 2001; 2010; Park et al., 2004; 2007; 2008; Fadnavis et al., 2014), and aerosols, forming the Asian tropopause aerosol layer (ATAL) (Vernier et al., 2011; 2015; Thomason and Vernier, 2013). This is due to persistent deep convection over heavily polluted regions, such as the Indian

subcontinent and south-east Asia, lifting pollutants from the boundary layer to the upper troposphere (UT), where the anticyclonic winds keep the air masses horizontally confined. The unique dynamical structure of the ASMA, with tropopause located at higher potential temperature than all its surroundings ($\theta > 380$ K), provides a potentially very efficient pathway for the transport of these pollutants into the lower stratosphere (LS). The transport can occur either vertically across the tropopause ("chimney model"), or isentropically across the horizontal boundaries of ASMA, hence bypassing the cold-point tropopause

("blower model") (Pan et al., 2016). Lagrangian trajectory calculations suggest that about half of the air mass in ASMA enters the stratosphere by the end of the ASM season (Garny and Randel, 2016). Lagrangian trajectories were also used to investigate the origin of the air masses in ASMA (Bergman et al., 2013; Vogel et al., 2015), yet this approach is limited by the convective



nature of the transport. Nevertheless, these studies are consistent with satellite observations (Fu et al., 2006), regional weather forecasting (Heath and Fuelberg, 2014) and global atmospheric circulation models (Fadnavis et al., 2013; Pan et al., 2016) in indicating the region of the southern slopes of the Himalayas (i.e. approximately latitudes 25-35°N south of the Tibetan plateau) as a hot-spot for the transport of boundary layer pollutants to the ASMA. Considering the recent rapid increase of pollutant

emissions from India (e.g. Krotkov et al., 2016) and south-east Asia, due to their fast economic growth, it is crucial for global chemistry climate models to properly represent the ASMA dynamics, thermodynamic structure and processes.

Up to date, most of the observational evidence regarding the chemical composition of the Asian UTLS is derived from satellite measurements, providing information with good regional and temporal coverage, but with limited vertical resolution. High vertical resolution is important to understand the physical boundaries that control the vertical distribution of chemical species,

and microphysical processes like the nucleation of cirrus clouds and aerosols. This requires accurate in-situ measurements in a notoriously hardly accessible region such as the UTLS, and balloon-borne measurements are particularly suited for this aim. In this article we present and discuss the data collected by the StratoClim balloon campaigns, carried out from two sites at the southern slopes of the Himalayas through the years 2016 and 2017. In total 63 balloon soundings were performed in Nainital (India) and Dhulikhel (Nepal) during three distinct periods of time, including two main monsoon-season campaigns (August

2016 in Nainital and July-August 2017 in Dhulikhel), and one brief post-monsoon campaign (November 2016 in Nainital). State-of-the-art instruments were used to measure vertical profiles of temperature, water vapor, ozone and aerosol backscatter, from the surface to the middle stratosphere. Here we first provide an overview of all measurements, showing mean profiles and their standard deviation ranges of natural variability for the different campaign periods, and then focus on analyzing the thermodynamic structure of the UTLS during the ASM season and how it relates with the distributions and variability of water

vapor, ozone and aerosols. One aim of this work is also to pave the way for ongoing more targeted modeling and intercomparison studies within StratoClim and other activities.

## 2. Campaign description, instruments and data processing

The measurements were performed in Nainital, Uttarakhand, India (29.35°N, 79.46°E: NT) and Dhulikhel, Nepal (27.62°N, 85.54°E: DK), hosted respectively by the Aryabhatta Research Institute of Observational Sciences (ARIES) and Kathmandu

University (KU). Both sites are located at the southern slopes of the Himalayan mountain range, at elevations of 1820 m (NT) and 1530 m (DK) above sea level. In this region, the terrain elevation increases steeply from sea-level heights of the Indo-Gangetic plane to the south, to elevations above 3000 m of the Tibetan plateau to the north, and strong orographic forcing induces persistent deep convection and heavy rainfall during the monsoon season (Vellore et al., 2016).

The measurements were conducted during three distinct periods of time, with two main monsoon campaigns, in NT in 2016

(2-31 August, 30 balloon soundings: $NT16_{AUG}$) and DK in 2017 (30 July - 12 August, 28 balloon soundings: DK17), and one shorter post-monsoon campaign in NT (2-8 November 2016, 5 balloon soundings: $NT16_{NOV}$). The frequency of the soundings and the composition of the payloads varied depending on meteorological conditions and on operational constraints. In the past,



attempts of conducting field campaigns in the ASM region were often hindered by operational and logistic limitations, which also affected our DK17 campaign resulting in a reduced measurement schedule (most notably, the number of backscatter measurements was limited: see Table 1). Nevertheless, important scientific data were collected. The DK17 campaign took place simultaneously with the StratoClim aircraft campaign, based in Kathmandu Airport (Nepal), which performed 8 scientific

5 flights with the high-altitude Geophysica-M55 research aircraft.

All soundings employed meteorological latex balloons (Totex, Japan) filled with hydrogen gas in order to ascend at a rate of about 5 m/s. Maximum burst altitude of these balloons is around 35 km, and more than 70% of our soundings reached at least 30 km (see Table S1 in supplementary material). A standard meteorological radiosonde was used to host additional instruments through its XDATA interface (Oelsner and Tietz, 2017), and to transmit the data of all instruments to the ground station at 1

10 Hz frequency. In particular, we used RS41-SGP (Vaisala, Finland) radiosondes (Vaisala, 2017), and the DigiCora MW41 sounding system (Vaisala, Finland) as ground station (Vaisala, 2014). Additional instruments employed were: Electrochemical Concentration Cell (ECC, manufacturer: En-Sci, USA) (Komhyr, 1969) for ozone ($O_3$) mixing ratio, Cryogenic Frost-point Hygrometer (CFH, En-Sci, USA) (Vömel et al., 2007; 2016) for water vapor ($H_2O$) mixing ratio, and Compact Optical Backscatter Aerosol Detector (COBALD, MyLab, Switzerland) for aerosol backscatter.

For the pressure ($p$) and temperature ($T$) measurements that we analyze in this work, uncertainties of RS41-SGP (hereafter: RS41) given by the manufacturer are 0.6/1 hPa (at pressures lower/higher than 100 hPa) and 0.3/0.4 K (at altitudes lower/higher than 16 km), respectively. The performances of ECC sondes have been assessed by many studies (e.g. Smit et al., 2007), and uncertainties are estimated as 5-10% in terms of $O_3$ partial pressure. CFH is a frost-point hygrometer based on the chilled-mirror principle with uncertainty on $H_2O$ mixing ratio lower than 10% up to 28 km altitude (Vömel et al., 2007). ECC and

CFH are regularly deployed in the ASM region since 2009 (Bian et al., 2012). COBALD is a detector for aerosol backscatter measurements at optical wavelengths of 455 nm (blue visible) and 940 nm (infrared) developed at ETH Zürich, downscaling the original backscatter sonde by Rosen and Kjome (1991) in weight and size. COBALD is able to detect cirrus clouds (e.g. Brabec et al., 2012) as well as aerosol layers, such as ATAL (Vernier et al., 2015). In addition, one RS92-SGP radiosonde (Vaisala, Finland) was added to almost all payloads for an intercomparison with the performances of RS41 (not discussed in

this paper). Finally, we note that for logistical constraints, the first two soundings in NT16$_{AUG}$ employed iMet-1-RSB radiosondes (InterMet, USA) (InterMet, 2006), offering the XDATA interface (Wendell and Jordan, 2016), instead of RS41, and SkySonde version 1.9 (Jordan and Hall, 2016) as data acquisition software. The number of deployments of each instrument during the different campaign periods are summarized in Table 1. A list of all the 63 soundings with date and time of launch, payload description, burst altitude and other notes is given in Table S1 in supplementary material.

In this study, we use the pressure measured by RS41 as the main vertical coordinate for all instruments. All variables are binned in pressure intervals of 1 hPa for $p > 300$ hPa, and 0.5 hPa for $p < 300$ hPa, yielding an improved signal-to-noise ratio and a dataset with consistent vertical levels. This binning corresponds to a vertical resolution of approximately 25 m in the UTLS. A quality check is performed for all instruments based on interpretation of their house-keeping data, and data points showing anomalous behavior are rejected (note that the number of malfunctioning events per instruments are noted in brackets



in Table 1). In this context, CFH deserves a special mention, since a behavior we named "CFH contamination" was observed in a significant number of cases. This consists in the drift towards too high frostpoint temperatures, hence too high water vapor mixing ratios, in the stratosphere (see Figure S1 in supplementary material) which we attribute to the deposition of supercooled water droplets onto the inner walls of the instrument's inlet tube while passing through mixed-phase clouds. This hypothesis is currently subject of a dedicated modeling study. To avoid such instrumental artifact, for the analysis in this paper we do not accept $H_2O$ mixing ratio measurements higher than 10 ppmv in the stratosphere, which are unrealistic, as well as all measurements at pressures below 20 hPa (see Section 3.3). Ice saturation ($S_{ice}$), i.e. relative humidity with respect to ice, is calculated using the frost-point temperature measured by CFH, air temperature by RS41, and the parameterization for saturation vapor pressure over ice by Murphy and Koop (2005). The COBALD data are expressed as backscatter ratio (BSR), i.e. the ratio of the total-to-molecular backscatter coefficient. This is calculated by dividing the total measured signal to its molecular contribution, which is computed from the atmospheric extinction according to Bucholtz (1995), and using air density derived from the measurements of temperature and pressure (see Cirisian et al., 2014). Uncertainty on COBALD BSR as inferred by this technique is estimated around 5% (Vernier et al., 2015). Finally, note that for conversion of pressure to geometric altitude ($z$), mean profiles of $p$ vs. $z$ measured by RS41 are shown in Figure S2 in supplementary material.

## 3. General overview of the data

Figure 1 shows mean profiles and standard deviations of temperature, $H_2O$ mixing ratio and $O_3$ mixing ratio calculated from all measurements performed during the three campaign periods, namely NT16$_{AUG}$ (blue), NT16$_{NOV}$ (green) and DK17 (red). Note that for improved recognition, we keep this color coding also in subsequent figures. Panels a-c show the entire measured profiles, while Panels d-f show zooms into the UTLS region. In this section we briefly discuss their main features. Aerosol backscatter measurements will be discussed in Section 6.

### 3.1 Troposphere

In the troposphere, seasonal variability of water vapor mixing ratio is the most evident feature, with differences up to a factor of 40 in the lower troposphere between the dry (November) and the ASM season (July-August). Massive latent heat release by condensation during the wet period, in contrast to dry conditions in winter, is reflected in different lower tropospheric lapse rates for the two seasons, with about 5.5 K/km (close to the moist adiabat) in July-August, and 8 K/km (close to the dry adiabat) in November. Lower tropospheric $O_3$ in NT16$_{AUG}$ compared to NT16$_{NOV}$ is likely due to enhanced washout of ozone precursor gases during the wet season. Higher $O_3$ in the lower troposphere in NT16$_{AUG}$ vs. DK17 might be due to photochemical smog transport from the New Delhi urban area and the highly populated Indo-Gangetic plane (e.g. Kumar et al., 2010). Additionally, in the upper troposphere, the same feature is likely also related to stronger convective activity occurring during DK17 compared to NT16$_{AUG}$ (discussed in Section 5.2).



### 3.2 Tropopause region

The structure of the tropopause region is very different during the three measurement periods. In contrast to the sharp cold-point tropopause (CPT) of the ASM season, typical of the tropics, the tropopause structure in November has more similarity to that in the mid-latitudes, with an almost isothermal layer above the lapse-rate tropopause (LRT), defined according to the

World Meteorological Organization (WMO, 1957). As a consequence, in $NT16_{AUG}$ and DK17 the LRT coincides with the CPT, whereas in $NT16_{NOV}$ the average LRT was found at 108 hPa and the CPT at 69.5 hPa, i.e. about 2.5 km apart. Interestingly, comparing the two ASM seasons datasets also reveals significant differences. The average CPT is 10 hPa higher (88 vs. 97.5 hPa, corresponding to about 600 m in altitude) and 5 K colder (-81.7 vs. -76.8 °C) in DK17 compared to $NT16_{AUG}$. Water vapor in the UTLS is minimum in $NT16_{NOV}$, with mixing ratios around 2.5 ppmv above the LRT. During the ASM, UTLS

$H_2O$ is higher, but different vertical distributions are observed. In $NT16_{AUG}$, $H_2O$ mixing ratio decreases monotonically with altitude, with mean value of 6.8 ppmv at the CPT. In DK17, $H_2O$ mixing ratio shows a minimum at the CPT (3.5 ppmv), and a local maximum in the LS (6 ppmv). Lower $O_3$ in the tropopaue region during the ASM than in November is likely related to the monsoonal deep convection, transporting ozone-poor air from the boundary layer to the UT. Mean altitude, pressure, potential temperature and temperature of the LRT and CPT for the three campaign periods are summarized in Table 2.

### 3.3 Stratosphere

Lower stratospheric temperatures (20-60 hPa) differ by about 2-4 K between November and July-August, which is consistent with the climatological annual cycle of stratospheric temperature (Randel et al., 2003). Stratospheric $H_2O$ mixing ratios are in the range of 4-6 ppmv up to 20 hPa, with a slight increase with altitude due to oxidation of methane. Above approximately 20 hPa (27 km), all CFH measurements show an unrealistic increase in $H_2O$ mixing ratio, which is a measurement artifact. At

such high altitudes and the corresponding dry conditions and low air densities, outgassing from the balloon skin and the payload train can play a significant role in contaminating the humidity measurements (Kräuchi et al., 2016). Hence, water vapor data in this range will not be considered in this analysis. Differences in stratospheric ozone between $NT16_{AUG}$ and DK17 are likely due to interannual variability.

### 4. Meteorological analysis

For relating the measurements to the large-scale atmospheric flow, here we analyze meteorological data from the European Center for Medium-range Weather Forecast (ECMWF) for the three campaign periods, focusing on the UTLS region.



## 4.1 Seasonal variability

Figure 2 illustrates the seasonal variability of the meteorological systems above the southern slopes of the Himalayas during NT16$_{AUG}$ (top row), NT16$_{NOV}$ (center), and DK17 (bottom row), in terms of latitude-pressure cross-sections (left column) and air mass backward trajectories (right column).

In particular, Figure 2 (left column) shows latitude-pressure cross sections of potential vorticity (PV), potential temperature ($\theta$), zonal wind speed and ice water content, for NT (longitude 80° N, Panels a, c) and DK (longitude 85°E, Panel e) from ECMWF analysis data on selected days. During the ASM season (Panels a, e), the subtropical westerly jet stream limits the ASMA on its northern flank around 40-45°N, and marks the border between a "tropical regime" to the south, including the ASMA, and the mid-latitudes to the north. The dynamical tropopause (i.e. the PV = 2 PVU surface) is found at about 100 hPa

inside the ASMA, while north of the jet it decreases steeply with altitude. After the end of the monsoon season, the subtropical westerly jet migrates southward to 30-35°N and intensifies in strength. Hence, in November (Panel c), NT is located at the boundary between the tropical and mid-latitude regimes, i.e. below the jet and the associated tropopause break, resulting in the quasi-isothermal lapse rate between the LRT and CPT discussed in Section 3.2. Additionally, the large standard deviation of temperature in the tropopause region in NT16$_{NOV}$ is likely related to varying meridional position of the jet during the measure-

ment period (see Figure 1, Panel d).

Figure 2 (right column) shows 2-weeks backward air mass trajectories calculated by the Lagrangian Analysis Tool (LAGRANTO) (Wernli and Davies, 1997), based on ERA-Interim re-analysis wind fields, and initialized at 100 hPa at the time of each sounding. During the ASM season (Panels b, f) the UTLS flow at the southern slopes of the Himalayas is easterly and follows the southern branch of the ASMA, transporting air which has been circulating around the anticyclone already for

several days. In contrast, in November (Panel d), the UTLS winds are westerly and follow the subtropical jet stream, carrying air from above the Saharan desert and air masses deflected from the equatorial easterlies. This is consistent with wind speed and direction measurements by RS41 shown in Figure S3 in supplementary material.

## 4.2 Interannual and regional variability

To assess whether the differences between the observations in NT16$_{AUG}$ and DK17 are caused mainly by geographic difference,

and associated different mesoscale weather features, or by interannual variability between the ASM 2016 and 2017 seasons, in Figure 3 we examine time series of UTLS temperature and H$_2$O mixing ratio from ECMWF analysis data for both stations and both campaign periods.

In August 2016 (Panels a-d), the UTLS was relatively warm at both locations, with CPT temperatures rarely below -80°C and DK slightly colder than NT (0.4 K on average at 100 hPa), and H$_2$O mixing ratio did never decrease below 4.5 ppmv at both

sites. The same day-to-day variability features are visible at both locations with a time shift of about 6-12 h, which is consistent with DK being systematically upstream of NT along the southern branch of the ASMA, and a wind speed of around 20 m/s in



the UTLS. In July-August 2017 (Panels e-h), $T$ and $H_2O$ values and features are similar to 2016 until 3 August. Then, a period characterized by extremely cold and dry tropopause starts in both NT and DK, peaking between 7-10 August with CPT temperatures colder than -83°C and $H_2O$ mixing ratios lower than 3 ppmv. The minima are slightly more pronounced in DK than in NT but are clearly correlated in time, suggesting that these features are related to a large-scale cooling and drying pattern

occurring in the ASMA. Interestingly, we also note that a layer of high $H_2O$ rises to high altitudes (70-85 hPa) after 3 August (Panel f), forming the local maximum above the CPT which we also find in our DK17 measurements (Figure 1, Panel e).

To quantify this resemblance, Figure 4 shows mean measured profiles of temperature (Panels a-b) and $H_2O$ mixing ratio (Panels d-e) in comparison with ECMWF operational data (analysis and forecast) for the NT16$_{AUG}$ and DK17 campaign periods. Good agreement in the location and magnitude of the CPT and of the main water vapor features, despite a slight moist bias of

the ECMWF data, suggests that the ECMWF model captures the main dynamical and microphysical features of the measurement periods. The $H_2O$ maximum above the CPT observed in DK17 and reproduced by the ECMWF model is remarkable and not in accordance with more typical climatological conditions observed during NT16$_{AUG}$. As will be discussed in Section 5, several features in our measurements suggest that the anomalous characteristics of the UTLS observed in DK17 are due to stronger convective activity compared to NT16$_{AUG}$.

In summary, here we argue that the differences between the NT16$_{AUG}$ and DK17 datasets are not due to local meteorological effects, which appear to have negligible impact on the UTLS temperature and water vapor at the two measurement sites. Rather, these differences are to be attributed to interannual variability, and in particular to anomalously strong convective activity occurring in the ASM after 3 August 2017 and persisting on a large scale.

## 4.3 Horizontal confinement in ASMA

In conclusion of the meteorological overview, we evaluate the vertical extent of the horizontal confinement effect of ASMA during the NT16$_{AUG}$ and DK17 campaign periods by means of air mass trajectories. For this, we calculate 2-weeks backward trajectories with LAGRANTO using ERA-Interim re-analysis wind fields, initialized between 40 and 150 hPa at the time of each balloon sounding in NT16$_{AUG}$ and DK17 (i.e. same as shown in Figure 2, Panels b, f for 100 hPa) and 6 h before and 6 h after each sounding. For each pressure level, we then calculate the "confined fraction" of trajectories, defined as the fraction

of trajectories which were already located inside the anticyclone 2 weeks before the measurements. The ASMA area is approximated as the box of 10-50°N latitude, 20-140°E longitude (shown by grey dashed lines in Figure 2, Panels b, f). The resulting confined fractions for NT16$_{AUG}$ and DK17 are plotted as a function of pressure in Figure 5. In both campaign periods, the confined fraction is high (above 60%) up to 70-80 hPa, while above it sharply decreases to zero. Therefore, we define the top of confinement (TOC) level of ASMA as the level of confined fraction = 50%, corresponding to 73 hPa in NT16$_{AUG}$ and

63.5 hPa in DK17. This level separates altitudes which are affected by the anticyclonic confinement of ASMA (below the TOC), from the confinement-free lower stratosphere above the ASMA (above TOC). Mean altitude and potential temperature of the TOC levels derived from the balloon measurements are given in Table 2. In Sections 5 and 6, we will show that the



TOC plays an important role in controlling the vertical distributions and variability of water vapor, ozone and aerosol backscatter in the Asian UTLS.

## 5. UTLS structure during the ASM season

In this section we focus on analyzing the UTLS structure of the NT16$_{AUG}$ and DK17 measurements. The NT16$_{NOV}$ measure-
5 ments will be discussed again in Section 6.

### 5.1 Thermodynamic structure

In the Brewer-Dobson circulation, tropical CPT temperature is thought to control the amount of water vapor entering the stratosphere (e.g. Mote et al., 1996). However, the thermodynamic transition between the troposphere and the stratosphere does not occur instantaneously, but over a layer of several kilometers in thickness, termed tropical tropopause layer (TTL),
and the properties of the TTL rather than CPT temperature alone control water vapor transport through the tropopause (High-wood and Hoskins, 1998; Holton and Gettelman, 2001; Fueglistaler et al., 2009). As discussed by Pan et al. (2014), different definitions of TTL are used in the literature. Gettelman and de F. Forster (2002) identify the TTL boundaries based on temperature profiles and derived lapse rates only, which is particularly suited for our measurements. In their definition, the upper boundary of the TTL is the CPT, and the lower boundary the lapse rate minimum (LRM) level, i.e. the point in altitude where
the change of potential temperature ($\theta$) with altitude ($z$) is minimum. This defines the TTL as the layer in which the temperature lapse rate switches from convectively-dominated, in the troposphere (small $d\theta/dz$, low stability), to radiatively-controlled in the stratosphere (high $d\theta/dz$, high stability) (Gettelman and de F. Forster, 2002). In addition, the LRM coincides with the mean convective outflow level (Gettelman and de F. Forster, 2002; Vömel et al. 2002; Paulik and Birner, 2012). Using this definition of TTL, Pan et al. (2014) showed that water vapor has low variability above the CPT, and ozone below the LRM.
Based on the similarity between the Asian tropopause region during the ASM season and that of the tropics (see Sections 3.2, 4.1), here we adopt this definition of TTL to study the thermal structure of the UTLS in our NT16$_{AUG}$ and DK17 datasets. However, being our measurement sites not tropical in a strict geographic sense (NT and DK are at latitudes 27-30°N), we refer to the TTL in this region and season as the Asian tropopause transition layer (ATTL, or Asian TTL).
Figure 6 shows mean profiles and standard deviations of $T$, $\theta$ and $d\theta/dz$ as a function of pressure for the two ASM season
datasets. As already noted in Section 3.2, the average CPT is higher in altitude by about 10 hPa and 5 K colder in DK17 than in NT16$_{AUG}$ (Panel a). Because of this large temperature difference, potential temperature levels in DK17 are shifted to lower pressures than in NT16$_{AUG}$ (Panel b), and the resulting difference between the two CPTs in isentropic coordinates is small (1.5 K), with both located slightly above the 380 K $\theta$-level. The average LRM is also found at higher altitude by roughly 10 hPa in DK17 compared to NT16$_{AUG}$ (Panel c), corresponding to a 2.5 K difference near the 360 K isentropic level. Higher LRM and
higher and colder CPT in DK17 suggest that, on average, convection reached higher altitudes in the ASM 2017 compared to



the year 2016, at least prior and during the time of our measurements. The exact ATTL boundaries in terms of pressure and average potential temperature are 180-97.5 hPa and 360-382 K for NT16$_{AUG}$, and 169.5-88 hPa, 362.5-383.5 K for DK17 (see Table 2: note that the potential temperature values are the average θ in the pressure levels where the LRM and CPT occur, and the data of individual soundings are not binned with respect to potential temperature). We also note that the measured profiles

of $d\theta/dz$ and mean LRM levels are in good agreement with ECMWF operational data (Figure 4, Panels c, f).

Figure 7 highlights the levels and layers used throughout this work (see also the summary in Figure 12 at the end of this paper). Based on the definitions of LRM, CPT and TOC (introduced in Section 4.3), hereafter we will refer to: UT as the region of altitudes below the LRM, Asian TTL (ATTL) as the region between LRM and CPT, Confined LS (CLS) as the region between CPT and TOC, and Free LS (FLS) as the region of altitudes above the TOC. In the following of this section, we use these

thermodynamically-significant levels and layers to study the vertical distribution and variability of water vapor, ozone and ice saturation.

## 5.2 Water vapor and ozone

To analyze the vertical distributions and variability of tracers in relation to the thermodynamic structure of the UTLS, we define for each balloon sounding the altitude relative to the CPT as a new vertical coordinate. Figure 7 shows mean profiles

and standard deviations of temperature, H$_2$O mixing ratio and O$_3$ mixing ratio in this new coordinate system. Besides the CPT (black dashed line), the average LRM and TOC levels are shown by dashed and dotted lines, respectively (blue for NT16$_{AUG}$, red for DK17). Water vapor mixing ratio in DK17 shows a minimum at the CPT and a local isolated maximum in the Confined LS (Panel b), centered about 1 km above the local CPT (i.e. not the average CPT, but evaluated for each profile individually). The H$_2$O minimum at the CPT is conceivably due to unusually high frequency of occurrence of ice clouds near the CPT in

DK17 (see Ice saturation discussion in Section 5.3), which deplete water vapor from the gas phase in favour of the condensed phase, hence resulting in a strongly dehydrated CPT. The H$_2$O maximum in the CLS is likely due to overshooting convective updrafts, injecting ice crystals directly above the CPT, which then evaporate at higher levels and hence hydrate the CLS. A minimum in O$_3$ mixing ratio slightly above the LRM is found in DK17 (Panel c), which is characteristic of very strong convection, quickly transporting ozone-poor air from the boundary layer to the TTL (Gettelman and de F. Forster, 2002; Vömel

et al. 2002; Paulik and Birner, 2012). The absence of this feature in NT16$_{AUG}$ corroborates the statement that convection was stronger during DK17 compared to the ASM 2016 season.

Figure 8 shows probability density functions (PDFs) of temperature (left column), H$_2$O mixing ratio (center) and O$_3$ mixing ratio (right column) calculated in the FLS (Panels a-c), CLS (Panels d-f), ATTL (Panels g-i) and UT (Panels j-l) regions for NT16$_{AUG}$ and DK17.

In both datasets, the PDFs of H$_2$O mixing ratio show enhanced water vapor in the Confined LS (Panel e) compared to the Free LS (Panel b). In particular, the PDFs in the CLS are skewed towards high values, similarly to what was found by aircraft



measurements in the tropics (Corti et al., 2008), and show higher frequency of occurrence of high $H_2O$ mixing ratios (approximately in the range 4.5-7 ppmv) than in the FLS. In contrast, PDFs in the FLS are narrower and show the expected distribution of background stratospheric $H_2O$ mixing ratios, with low variability and rarely showing values larger than 5 ppmv. Higher $H_2O$ mixing ratios in the CLS compared to the FLS in DK17 are obviously related to the previously discussed isolated maxi-
mum, yet enhanced frequency of occurrence of high $H_2O$ mixing ratios is also observed in $NT16_{AUG}$ despite no local maximum was found in this dataset. We argue that this enhancement is due to the anticyclonic confinement effect of ASMA, which keeps the moist convective outflow horizontally confined and thereby increases the frequency of occurrence of high $H_2O$ mixing ratios above the CPT in ASMA. To quantify this enhancement, we consider the mean values of $H_2O$ mixing ratio in the CLS and FLS, which are respectively 4.93 ppmv, 4.42 ppmv in $NT16_{AUG}$, and 5.05 ppmv, 4.29 ppmv in DK17. This means that, on
average, 0.49 ppmv ($NT16_{AUG}$) and 0.76 ppmv (DK17) more $H_2O$ are found in the Confined LS inside the ASMA, compared to background stratospheric water vapor in the Free LS above the ASMA, i.e. that $H_2O$ mixing ratio is enhanced by 11-18% in the lowermost stratosphere due to the confinement effect of ASMA.

The PDFs of $O_3$ mixing ratio in the CLS, ATTL and UT (Panels f, i, l) show higher frequency of occurrence of low ozone values in DK17 compared to $NT16_{AUG}$, again suggesting stronger convective activity in the ASM 2017 than in 2016. Ozone
depletion extending into the Confined LS confirms that the influence of deep convection on the composition of the UTLS persists up to altitudes well above the CPT, due to the horizontal confinement in ASMA. The fact that this feature vanishes in the Free LS (Panel c), suggests that our trajectory-based definition of TOC is a good measure of the topmost extent of the anticyclonic confinement.

These evidence show that, in the ASM system, the top height of the horizontal confinement in ASMA controls the distribution
and variability of water vapor and ozone in the lowermost stratosphere, rather than CPT height as it is in the conventional understanding of the TTL.

### 5.3 Ice saturation

Figure 9 shows mean profiles and standard deviations of ice saturation (Panel a), histograms of supersaturated fraction (Panel b), and PDFs of ice saturation calculated for the UT, ATTL and CLS regions (Panels c-e). As a result of significantly colder
temperatures (Figure 7, Panel a), much higher and more persistent ice saturations were measured in DK17 than in $NT16_{AUG}$ throughout the entire ATTL. In both datasets, ice saturation is higher in the ATTL compared to the UT, and in DK17 it shows a pronounced maximum with average supersaturated conditions (i.e. more than 50% of the measurements reach $S_{ice} > 1$) in the 1.5 km directly below the CPT. In contrast, the supersaturated fraction is 10-15% in $NT16_{AUG}$ over the same range of altitude. The ATTL ice saturations and supersaturated fractions of $NT16_{AUG}$ are comparable with previous measurements in Lhasa and
Kunming, China between 2009-2012 (Bian et al., 2012) while the measurements in DK17 range significantly higher, confirming that the ASM 2017 season was indeed extraordinary in terms of strong convection, and also suggesting particularly high frequency of occurrence of cirrus clouds in the ASMA 2017. Interestingly, we also note that ice supersaturations in DK17





extend frequently into the CLS, with about 30% supersaturated fraction in the first 500 m above the CPT (Figure 9, Panels b, c). This implies that, in overshooting convective updrafts, ice crystals can regularly penetrate the CPT as condensed phase (e.g. see Figure 10, Panel f) and evaporate at higher levels in the Confined LS, similarly but likely more frequently than was found in the tropics (Corti et al., 2008). We consider this to be the reason for the isolated $H_2O$ maximum above the CPT

observed in DK17 (Figure 7, Panel b), which is, to our knowledge, unparalleled in the tropics.

## 6. Aerosol and cloud backscatter

Finally, here we analyze the aerosol and cloud backscatter measurements by COBALD, which have not been discussed so far. COBALD was originally designed and used to investigate the properties of ice cloud, including cirrus (e.g. Brabec et al., 2012; Cirisian et al., 2014) and polar stratospheric clouds (e.g. Engel et al., 2014), but as already mentioned, recent measurements

from Lhasa, China were also used for in-situ detection of the ATAL aerosols and validation of the Cloud-Aerosol Lidar and Infrared Pathfinder Satellite Observation (CALIPSO) satellite retrieval (Vernier et al., 2015). In this section we address both aspects.

Since the BSR of aerosol droplets is 1-2 orders of magnitude smaller than that of cirrus clouds, the characterization of ATAL requires cloud-filtering techniques to eliminate in-cloud measurements, and therefore a large dataset for a statistically-signifi-

cant evaluation (e.g. 18 soundings are used in Vernier et al., 2015). We performed 17 COBALD soundings in NT16$_{AUG}$, but due to logistical constraints only 3 could be realized during the DK17 campaign, which furthermore mostly sampled cloudy conditions in the ATTL. For this reason, a clear-sky aerosol BSR profile cannot be established from the DK17 dataset. On the other hand, the 3 COBALD soundings available from NT16$_{NOV}$ are almost fully clear-sky measurements and therefore allow to calculate a clear-sky aerosol BSR profile, providing a useful reference state of background aerosols (i.e. without ASMA

confinement) for comparison with NT16$_{AUG}$. In the following, we first provide an overview of the main characteristics of the observed cirrus clouds, and then detail the cloud-filtering technique and ATAL detection during the year 2016.

## 6.1 Cirrus clouds

Figure 10 shows individual soundings as examples of thin cirrus clouds observed in the ATTL during the NT16$_{AUG}$ (Panels a-d) and DK17 (Panels e-f) campaigns. Along with the temperature and ice saturation profiles, we show BSR at 455 nm ($BSR_{455}$),

BSR at 940 nm ($BSR_{940}$), and color index (CI). Color index is defined as the 940-to-455 nm ratio of the aerosol component of BSR, i.e. CI = $(BSR_{940} - 1)/(BSR_{455} - 1)$. CI has the property of being independent of number density, hence it is a useful indicator of particle size (e.g. Cirisian et al., 2014) as long as particles are sufficiently small, so that Mie scattering oscillations are avoided. Based on the size-dependence of CI, considerations on the typical size range of ice crystals and aerosol droplets, and the evaluation of ice saturation measurements by CFH, a threshold of CI = 7 was empirically developed to discriminate

in-cloud (CI > 7) from clear-sky measurements (CI < 7) (Vernier et al., 2015). This helps discerning the BSR features in Figure



10 as either ice cloud or aerosol signal, and is also used as a threshold for cloud-filtering. For example, in NT004 (Panel a), the sharp feature at 145 hPa with CI ≈ 10 is likely an ice cloud (note the concomitant ice supersaturation above the thin cloud layer, suggesting sedimentation), while the broad enhancement in BSR between 95-140 hPa without CI enhancement is the signal of ATAL. The main common characteristics of the cirrus clouds in Figure 10 is their very small spatial and optical

thickness, with $BSR_{940} < 20$, while much larger values ($BSR_{940} \gg 100$) are expected in homogeneously-nucleated cirrus clouds, as often observed in the midlatitudes (e.g. Brabec et al., 2012; Cirisian et al., 2014), and as also evidenced by the thick outflow cirrus below 120 hPa in DK002 (Panel f). Low BSR indicates low ice crystal number densities, suggesting that these clouds are most likely formed by heterogeneous nucleation on solid ice nuclei, rather than by homogeneous freezing of sulfate aerosol liquid droplets. This hypothesis is currently being investigated by a dedicated microphysical modeling study. Similarly thin

cirrus clouds were observed in about half of the COBALD soundings in NT16$_{AUG}$ (9 out of 17) therefore they occur very frequently in the ASMA, and they were often found embedded within the ATAL, as shown in Figure 10 (Panels a, e).

## 6.2 ATAL in the ASM 2016

Figure 11 shows all clear-sky (i.e. aerosol only) $BSR_{455}$ data points and mean profiles from the NT16$_{AUG}$ and NT16$_{NOV}$ datasets. Similarly to Vernier et al. (2015), the cloud-filtering criterion we applied consists of three thresholds from two independent

measurements, namely: $BSR_{940} < 2.5$ and CI < 7 from COBALD, and $S_{ice} < 0.7$ from CFH. Only data points which simultaneously fulfill all the three criteria above are classified as clear-sky and shown in Figure 11. The cloud-filtering method is illustrated by a scatter plot of $BSR_{940}$ vs. CI shown in Figure S4 in supplementary material.

In NT16$_{AUG}$, clear-sky $BSR_{455}$ enhancement starting approximately at the LRM (180 hPa) and extending up to the TOC (73 hPa) is the signature of ATAL (Figure 10), showing intensity and vertical extent comparable to those derived by CALIPSO

satellite retrievals and previous COBALD soundings (Vernier et al., 2015). The aerosol enhancement covers both the Asian TTL and the Confined LS with maximum $BSR_{455}$ at the CPT, suggesting that the LRM (i.e. the mean convective outflow level) is also the onset of the horizontal confinement in ASMA, and the CPT its level of maximum strength. Above the CPT, the $BSR_{455}$ enhancement gradually faints with altitude until the TOC, as the horizontal confinement effect by ASMA vanishes (see Figure 5). Above the TOC, the ATAL signal merges with the Junge layer of stratospheric aerosols, which extends into the Free

LS up to about 10 hPa. As expected, the aerosol enhancement is absent in NT16$_{NOV}$, showing that ATAL does not outlive the break-up of the anticyclonic vortex at the end of the ASM season.

## 7. Discussion and conclusions

We have analyzed 63 balloon measurements of temperature, water vapor, ozone and aerosol backscatter, collected in the region of the southern slopes of the Himalayas during the years 2016-2017. The NT16$_{AUG}$ campaign took place over one month (2-

31 August) with average frequency of one balloon sounding per day, hence covering roughly one third of the ASM 2016



season (June-August), characterized by typical climatological conditions. The DK17 campaign took place over two weeks (29 July-12 August) with average frequency of two balloon soundings per day, coincident with a period of remarkably anomalous characteristics of the UTLS. The measurements in NT16$_{NOV}$ span an even shorter time period (5 days), but provide a useful reference state for seasonal variability considerations.

5    The structure of the tropopause region at the southern slopes of the Himalayas exhibits strong seasonal variability, with typically tropical characteristics (sharp CPT) during the ASM season due to deep convection, and "mid-latitude" features (separated LRT and CPT) during the dry season. Therefore, for the analysis of our monsoon-season measurements, we adopt the concept of tropical tropopause layer (TTL) and define the region of altitudes between the LRM and the CPT as the Asian Tropopause Transition Layer (ATTL, or Asian TTL). Further, based on air mass trajectories, we define the top of confinement (TOC) level of ASMA, which separates the LS into a Confined LS (CLS), below the TOC and above the CPT, and a Free LS (FLS), above the TOC. These thermodynamically significant levels and layers, qualitatively illustrated in Figure 12, provide physically-meaningful boundaries for the analysis of the vertical distribution and variability of water vapor, ozone and aerosols in the UTLS.

Figure 13 summarizes mean profiles of the UTLS thermal structure and chemical composition measured during NT16$_{AUG}$ (top
panel) and DK17 (bottom panel). During both the ASM season campaigns, the isentropic level of the LRM ($\theta = 362\text{-}364$ K) was higher than in previous measurements in the tropical Western Pacific and Central America ($\theta \approx 345$ K) (Gettelman and de F. Forster, 2002; Pan et al., 2014) and in the Tibetan plateau (355-360 K) (Bian et al., 2012), suggesting that convection is very deep-penetrating at the southern slopes of the Himalayas. The CPT ($\theta = 382\text{-}384$ K) was also higher than at tropical sites (375 K) (Gettelman and de F. Forster, 2002; Pan et al., 2014) but lower than above the Tibetan plateau (390 K) (Bian et al.,
2012), possibly suggesting an orographic influence. It is also interesting to notice that, in both datasets, the TOC coincides with a change in slope of $d\theta/dz$. Due to stronger convection, both the LRM and CPT are found at lower pressures in DK17 compared to NT16$_{AUG}$ (by about 10 hPa), but due to concomitant colder temperatures in DK17, the shift in potential temperature space is small (around 2 K). Ozone mixing ratio in DK17 shows a slight minimum above the LRM, which indicates strong convection (Vömel et al., 2002; Paulik and Birner, 2012) and was not observed in NT16$_{AUG}$. In both datasets, water vapor is
higher in the Confined LS compared to the Free LS (see also Figure 8), and in DK17 the H$_2$O mean profile shows a local maximum about 1 km above the CPT. This feature contradicts the common conceptual understanding of TTL, which claims no water vapor variability above the CPT (Pan et al., 2014), and we suggest this is due to overshooting convection injecting condensed-phase ice directly above the CPT, which subsequently evaporate and hydrate the CLS. Ice saturation is minimum at the LRM and increases in the ATTL, similarly to as in the tropics (Vömel et al., 2002). Due to extremely cold temperatures,
average $S_{ice}$ in DK17 is remarkably higher than in NT16$_{AUG}$, as well as than in previous measurements from the Tibetan plateau (Bian et al., 2012). Aerosol backscatter enhancement of the ATAL was detected in NT16$_{AUG}$ with similar vertical extent and BSR intensity as in previous measurements from Lhasa, China (Vernier et al., 2015). On average, ATAL extends from the LRM to the TOC with maximum backscatter at the CPT, suggesting that the mean convective outflow level (i.e. the LRM) is also the onset of the horizontal confinement in ASMA, CPT its level of maximum strength, and TOC its topmost extent.



In conclusion, our analysis shows that the composition of the tropopause transition layer and the lowermost stratosphere within the ASM anticyclone are influenced by deep convection up to 1.5-2 km above the cold-point tropopause, due to the horizontal confinement effect of ASMA. Hence, the vertical distributions and variability of water vapor, ozone and aerosols in this region are controlled by the topmost extent of the anticyclonic confinement, rather than the altitude of the cold-point tropopause, as it is in the conventional understanding of the "generic" TTL. This is shown by enhanced water vapor and aerosol backscatter found in the confined lower stratosphere, in contrast to typical background stratospheric values observed above the top of confinement. These evidence strongly suggest that the ASM anticyclone contributes to moistening the global stratosphere and to increase its aerosol burden. For our two monsoon-season campaigns in 2016-2017, the water vapor enhancement is quantified respectively as 11-18% (0.5-0.75 ppmv), in terms of average $H_2O$ mixing ratio difference between the confined lower stratosphere (i.e. above the cold-point and inside the ASM anticyclone) and the background stratospheric $H_2O$ measured in the free lower stratosphere above the anticyclone.

Our approach based on significant thermodynamic levels, rather than fixed pressure or altitude stacks, provides physically-meaningful diagnostics for the comparison of our in-situ measurements with global climate model outputs. As often mentioned throughout the paper, a wide range of modeling, interpretation and intercomparison studies are ongoing, aiming to explore the different insights offered by this dataset. These investigations include microphysical model simulations, instrumental studies, intercomparisons with other in-situ measurements (including the airborne Geophysica-M55 measurements during the Strato-Clim 2017 campaign) and with different global modeling products. Further ongoing studies address the interactions of large-scale dynamics with the thermal structure of the UTLS and linkages to monsoon precipitation, evidence of overshooting convection and Lagrangian air mass origin analysis.

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



**Acknowledgements**

The research leading to these results has received funding from the European Community's Seventh Framework Programme (FP7/2007 - 2013) under grant agreement n° 603557 and the Swiss National Science Foundation in project n° 200021-117987. The use of ECMWF operational and ERA-Interim data is gratefully acknowledged. Support from the Director ARIES and the ISRO ATCTM project is highly acknowledged for the observations at Nainital. The author Simone Brunamonti thanks Dr. Federico Fierli and Dr. Laura Pan for inspiring discussion.



| Station | Time period | RS41-SGP (*) | ECC | CFH | COBALD | Early burst |
|---------|-------------|--------------|-----|-----|--------|-------------|
| NT | 2-31 Aug 2016 | 28 + 2* (0) | 24 (2) | 27 (1, 5) | 17 (0) | 4 |
| NT | 2-8 Nov 2016 | 5 (0) | 5 (0) | 5 (0, 1) | 3 (1) | 0 |
| DK | 30 Jul-12 Aug 2017 | 28 (0) | 12 (2) | 11 (0, 4) | 3 (0) | 5 |
| **NT+DK** | **Total** | **61 + 2* (0)** | **41 (4)** | **43 (1, 10)** | **23 (1)** | **9** |

**Table 1. Number of soundings performed for each instrument, station and campaign period. In parentheses: number of soundings with instrumental malfunctionings (for CFH: number of failures and contamination events, respectively). Early burst is defined as burst altitude < 25 km. (*) Note that iMet-1-RSB radiosondes were used for the first two soundings in NT16_AUG instead of RS41-SGP.**



| | NT16$_{AUG}$ | | | | NT16$_{NOV}$ | | | | DK17 | | | |
|---|---|---|---|---|---|---|---|---|---|---|---|---|
| | $z$ (km) | $p$ (hPa) | $\theta$ (K) | $T$ (°C) | $z$ (km) | $p$ (hPa) | $\theta$ (K) | $T$ (°C) | $z$ (km) | $p$ (hPa) | $\theta$ (K) | $T$ (°C) |
| **LRM** | 13.3 | 180 | 360 | -52.7 | 10.5 | 260 | 337.5 | -43.6 | 13.7 | 169.5 | 362.5 | -55 |
| **LRT** | 17.0 | 97.5 | 382 | -76.8 | 16.0 | 108 | 378 | -73.2 | 17.6 | 88 | 383.5 | -81.7 |
| **CPT** | 17.0 | 97.5 | 382 | -76.8 | 18.5 | 69.5 | 424 | -75.3 | 17.6 | 88 | 383.5 | -81.7 |
| **TOC** | 18.6 | 73 | 421.5 | -73.7 | N.A. | N.A. | N.A. | N.A. | 19.5 | 63.5 | 441 | -72.7 |

**Table 2.** Mean values of altitude ($z$), pressure ($p$), potential temperature ($\theta$) and temperature (T) of the lapse rate minimum (LRM), lapse rate tropopause (LRT), cold-point tropopause (CPT) and top of confinement (TOC) levels during the three campaign periods, NT16$_{AUG}$, NT16$_{NOV}$ and DK17. Note that for NT16$_{NOV}$, the definition of TOC is not applicable (N.A.).





**Figure 1. Mean profiles (solid lines) and standard deviations (color shading) of all measurements of temperature from RS41 (Panels a, d), H₂O mixing ratio from CFH (b, e), and O₃ mixing ratio from ECC (c, f) as a function of pressure, for NT16_AUG (blue), NT16_NOV (green) and DK17 (red). Horizontal lines indicate the pressure levels of the average cold-point tropopause (CPT, dashed) for the different datasets, and the lapse-rate tropopause (LRT, dotted) for NT16_NOV. Upper row (a-c): measured profiles from the surface to 10 hPa. Bottom row (d-f): zoom into the tropopause region (40-180 hPa).**





**Figure 2. Left column: latitude-pressure cross-sections along longitude 80°E (Panels a,c) and 85°E (e), of: potential vorticity (color scale), potential temperature (black contours, in K), zonal wind speed (blue contours, m/s) and ice water content (white contours, 8**



steps, log-scale between 0.1-1000 ppmm) from ECMWF operational analysis data (horizontal resolution: 0.125°, vertical resolution: L137) for: 8 August 2016 at 00 UTC (a), 11 November 2016 00 UTC (c), and 5 August 2017, 12 UTC (e). Black dashed lines show the latitude of NT (a, c) and DK (e). Right column: 2-weeks LAGRANTO backward trajectories along ERA-Interim wind fields (1°, L60), initialized at 100 hPa at the time of each balloon sounding in NT16$_{AUG}$ (Panel b), NT16$_{NOV}$ (c), and DK17 (f). Note that in Panel c (NT16$_{NOV}$), trajectories stared 6 h before and 6 h after each sounding are also included.





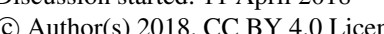

**Figure 3. Time series of temperature (left column) and water vapor mixing ratio (right column) as a function of pressure, from ECMWF operational analysis data (6-hourly, horizontal resolution: 0.125°, vertical resolution: L137) for the locations of NT (Panels c, d, g, h) and DK (a, b, e, f) during 1-31 August 2016 (a-d) and 20 July-21 August 2017 (e-h).**





**Figure 4. Comparison of temperature (Panels a, d), water vapor mixing ratio (b, e) and *dθ/dz* (c, f) mean profiles (solid lines) and standard deviations (color shading) of the NT16$_{AUG}$ (blue) and DK17 (red) campaigns, with ECMWF operational data (mean profiles and standard deviation bars, purple) for the periods: 1-31 August 2016 (a-c), 29 July-13 August 2017 (d, f) and 3-13 August 2017 (e). Dashed lines show the mean CPT (Panels a, d) and LRM (c, f) in the different datasets. ECMWF operational data consists of analysis and forecast (horizontal resolution: 0.125°, vertical resolution: L137) combined by taking analysis data files (6-hourly) for every**



time step at 00, 06, 12 and 18 UTC, and forecast data files (1-hourly) at all other time steps, from the most recent available forecast run (i.e. for every day, forecast data files initialized at 00 UTC are used for 01-11 UTC, and forecast initialized at 12 UTC for 13-23 UTC). Note that different periods are selected for Panels d, f and e because, while the DK17 campaign started on 29 July 2017, CFH water vapor measurements only started on 3 August 2017 (see Table S1 in supplementary material).





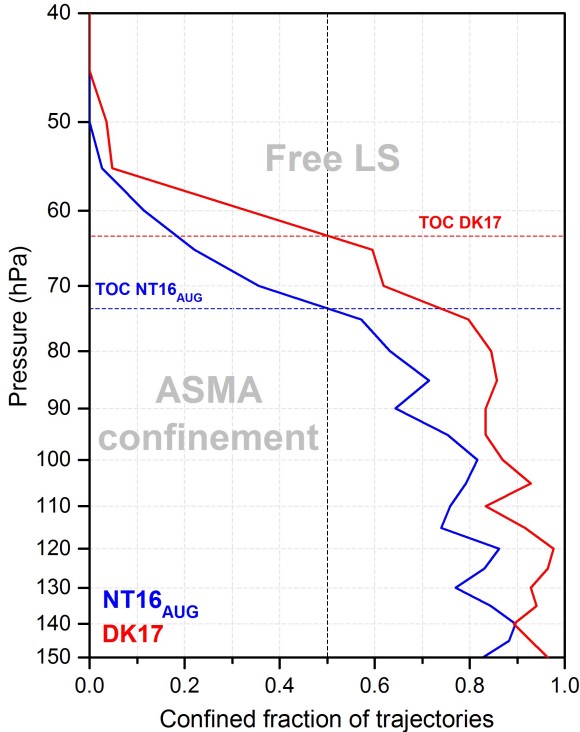

**Figure 5. Confined fraction of trajectories as a function of pressure (defined by 2-weeks backward trajectories, as described in Section 4.3) for the NT16$_{AUG}$ (blue) and DK17 (red) campaign periods. Dashed lines mark the TOC level for NT16 (blue) and DK17 (red).**



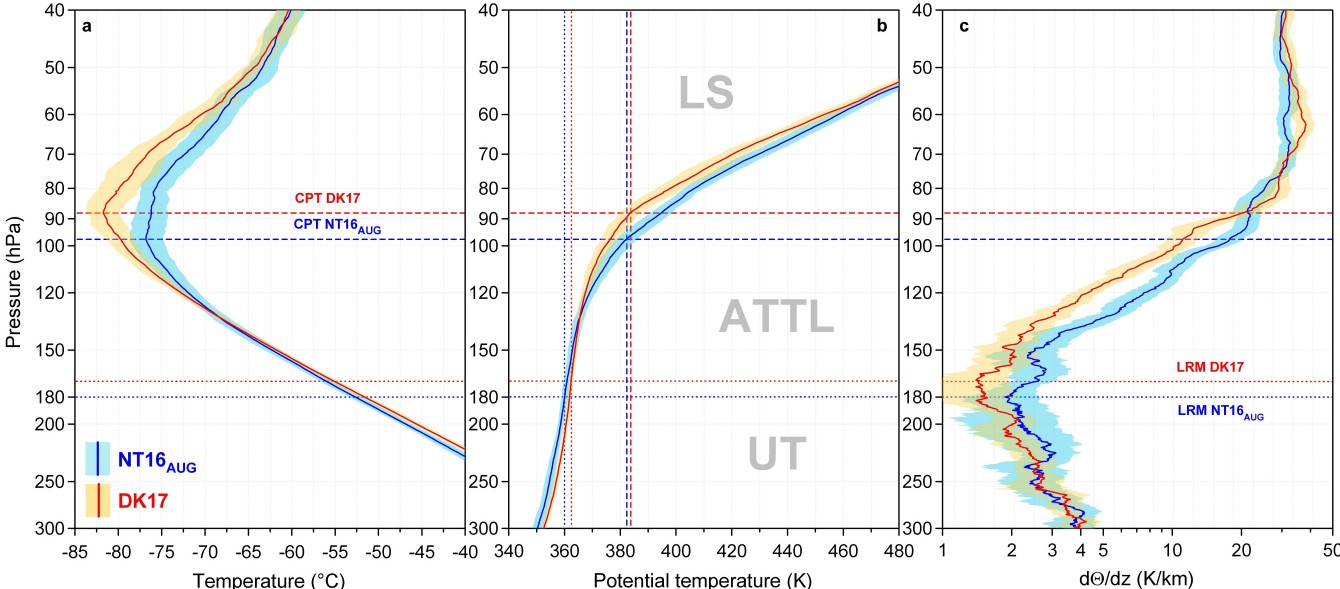

**Figure 6. Mean profiles (solid lines) and standard deviations (color shading) of temperature (Panel a), potential temperature (b) and $d\theta/dz$ (c), as a function of pressure, for NT16$_{AUG}$ (blue) and DK17 (red). Horizontal lines show the mean CPT (dashed) and LRM (dotted) for NT16 (blue), and DK17 (red). The region of the Asian tropopause transition layer (ATTL) defined in Section 5.1 is also highlighted. Note that mean profiles and standard deviations of $d\theta/dz$ c were smoothed with a $\pm 5$ hPa (about 250 m) moving average to reduce the noise in the GPS geometric altitude measurement by RS41.**



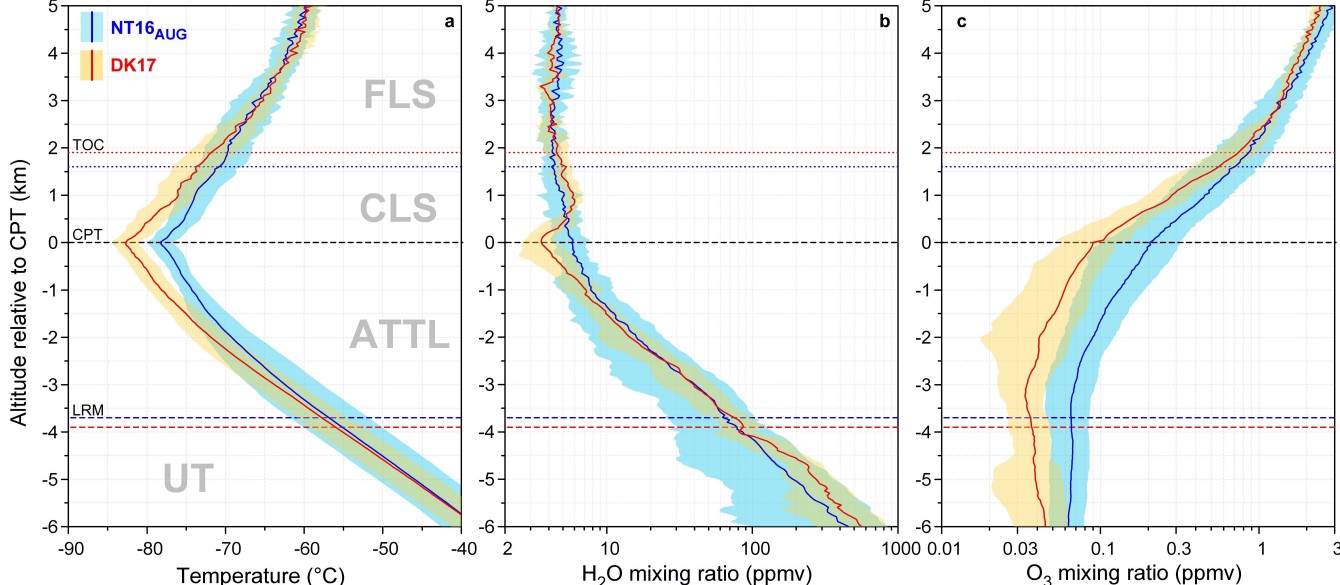

**Figure 7. Mean profiles (solid lines) and standard deviations (color shading) of temperature (Panel a), H$_2$O mixing ratio (b) and O$_3$ mixing ratio (c) as a function of altitude relative to CPT, for NT16$_{AUG}$ (blue) and DK17 (red). Dashed lines show the CPT (black) and the average LRM levels for NT16 (blue), and DK17 (red). Dotted lines show the TOC levels for NT16 (blue), and DK17 (red). The four layers defined in Section 5.1 (UT, ATTL, CLS, FLS) also highlighted.**



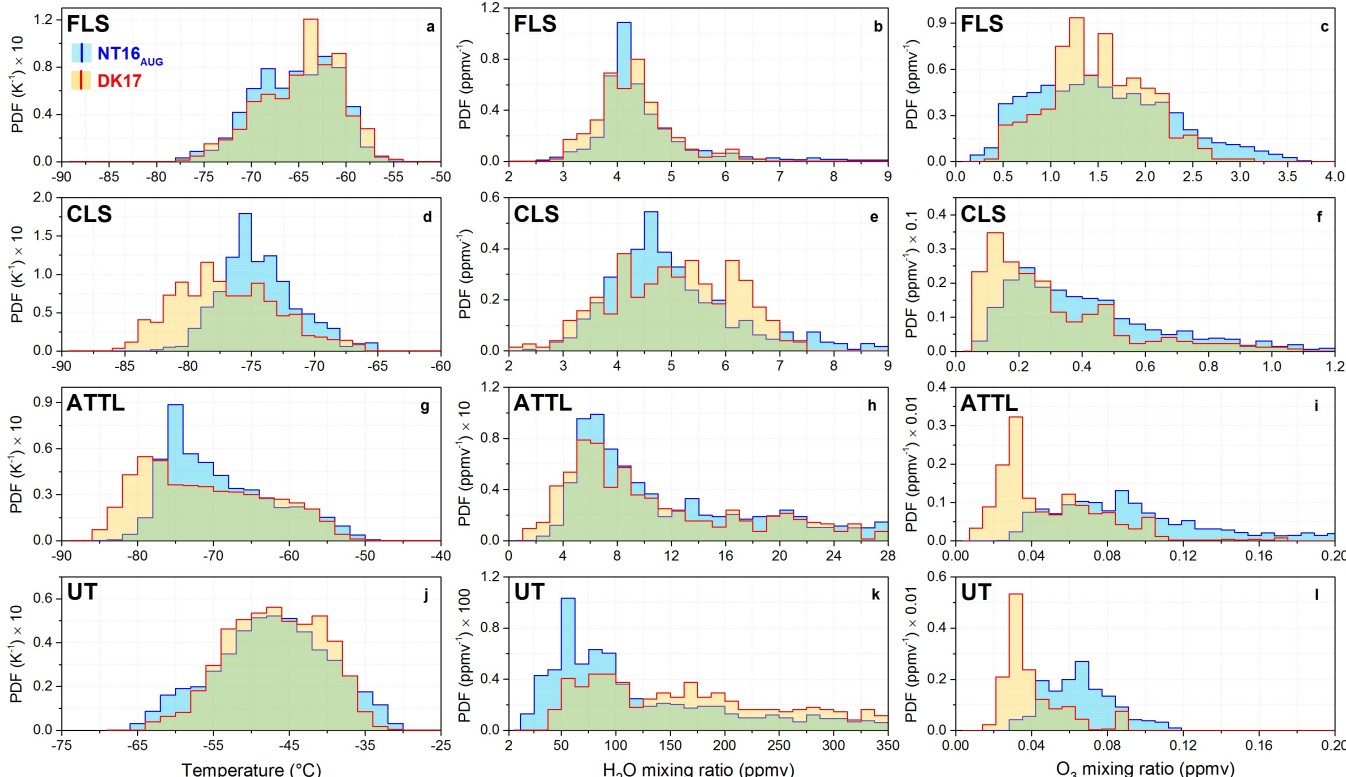

**Figure 8. Probability density functions (PDFs) of temperature (left column), $H_2O$ mixing ratio (center column) and $O_3$ mixing ratio (right column), calculated in the regions of Free LS (FLS, Panels a-c), Confined LS (CLS, d-f), Asian TTL (ATTL, g-h) and upper troposphere (UT, Panels j-l) as defined in Section 5.1, for NT16$_{AUG}$ (blue) and DK17 (red). PDFs of the UT region are calculated for altitudes between LRM and CPT - 6 km, while for the FLS between CPT and CPT + 5 km (i.e. the altitude range shown in Figure 7).**



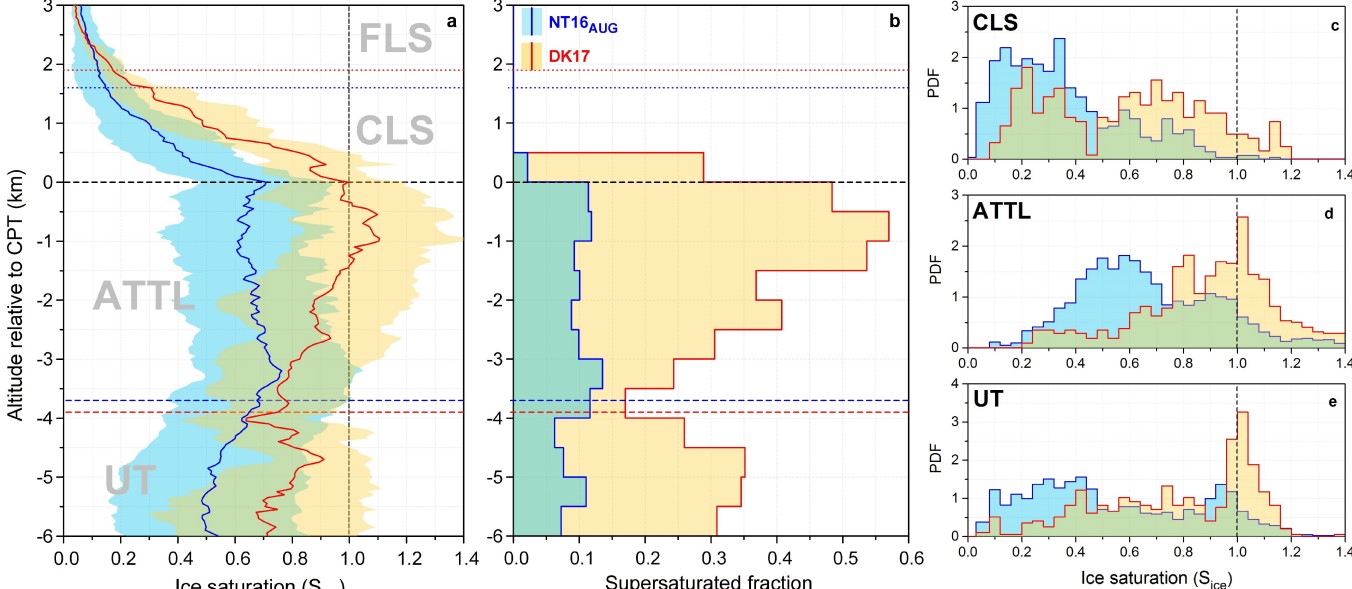

**Figure 9. Panel (a): mean profiles and standard deviation of ice saturation ($S_{ice}$) as a function of altitude relative to CPT, for NT16$_{AUG}$ (blue) and DK17 (red). Panel (b): supersaturated fraction (i.e. fraction of measurements with $S_{ice} > 1$) as a function of altitude relative to CPT. Panels (c, d, e): PDFs of ice saturation in the CLS, ATTL and UT regions, respectively.**

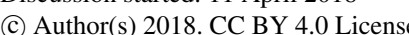



**Figure 10. Examples of thin cirrus clouds measured in the ATTL during four individual soundings of the NT16$_{AUG}$ campaign (Panels a-d) and two soundings of the DK17 campaign (e, f). Solid lines show temperature (black), ice saturation (light blue), BSR at 455 nm (blue), BSR at 940 nm (red) and Color index (green). Sounding identification numbers are noted in black on each panel (for date, time and payload of each sounding, see Table S1 in supplementary material). Vertical dashed lines mark the $S_{ice}$ = 1 (light blue) and Color index = 7 (green) thresholds.**




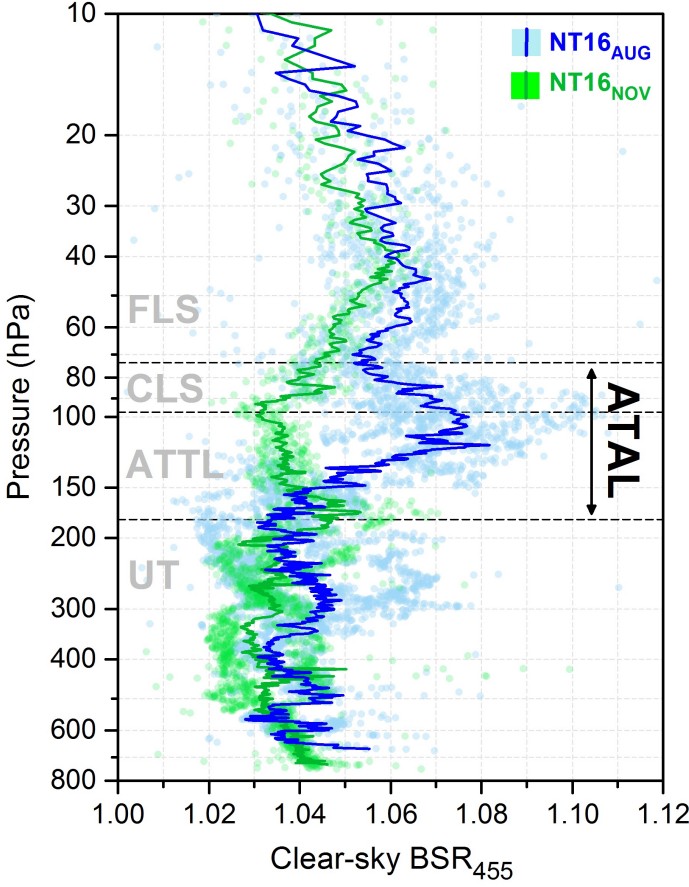

**Figure 11.** All clear-sky (i.e. aerosol-only) data points (dots) and mean profiles (solid lines) of $BSR_{455}$ as a function of pressure, for $NT16_{AUG}$ (blue) and $NT16_{NOV}$ (green). Black dashed lines show the mean LRM, CPT and TOC levels for $NT16_{AUG}$.





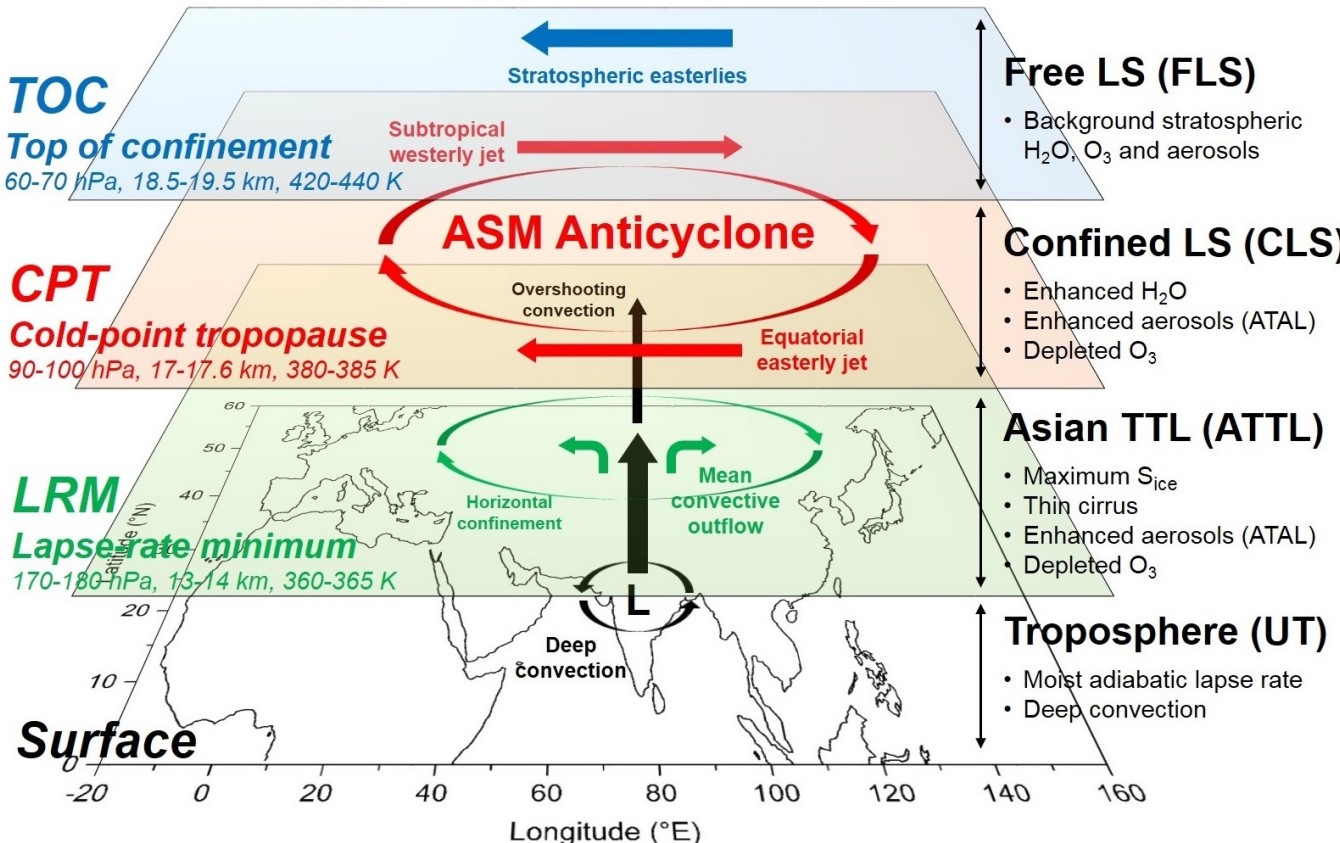

**Figure 12. Schematics of the vertical structure and main observed features of the UTLS above the southern slopes of the Himalayas. The Asian summer monsoon anticyclone (ASMA) consists of two layers, the Asian tropopause transition layer (ATTL) and the confined lower stratosphere (CLS). These layers are confined by three levels: the lapse rate minimum (LRM, green surface), the cold-point tropopause (CPT, red) and the top of confinement (TOC, blue).**

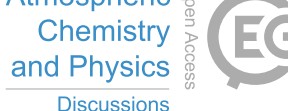



**Figure 13. Mean measured profiles of temperature (red),** *dθ/dz* **(purple), H₂O mixing ratio (blue), O₃ mixing ratio (orange), ice saturation (cyan) and clear-sky aerosol** *BSR₄₅₅* **(black) as a function of pressure (left axis, light grey dashed lines) for NT16ₐᵤG (top panel) and DK17 (bottom panel). Average potential temperature leves are shown on the right axis and marked by green dashed**



lines. Note that the pressure scale is the same for the two panels, and the potential temperature levels vary according to the measurements. The average CPT, LRM and TOC levels are marked by black solid lines. The Asian TTL (ATTL) and Confined LS (CLS) layers are marked by light green and orange shading, respectively. Note that mean profiles of $d\theta/dz$, ice saturation and clear-sky aerosol $BSR_{455}$ are smoothed with a ± 5 hPa (about 250 m) moving average (same as in Figure 6, Panel c).