# Peer review of "Balloon-borne measurements of temperature, water vapor, ozone and aerosol backscatter at the southern slopes of the Himalayas during StratoClim 2016-2017"

_Atmospheric Chemistry and Physics, 2018_

## Referee Comment (RC1) · Anonymous Referee #1 · 4 May 2018

General Comments

This manuscript presents the data and analyses of balloon-borne measurements from Northern India and on the southern slope of the Tibetan plateau during two Asian summer monsoon (ASM) seasons. The high vertical resolution profiles of temperature, water vapor, ozone, as well as the cirrus clouds and aerosol information are analyzed together to characterize the region's UTLS thermal and dynamical structure, transport characteristics, in particular the transport of water vapor into the stratosphere and the presence of the Asian tropopause aerosol layer (ATAL). This work is part of a larger

project involving the airborne campaign StratoClim. The data and the analyses are well documented in the manuscript. The result contributes important new information to the larger picture of ASM UTLS transport. The work is high quality and fits the scope of ACP well. I have a number of suggestions for improving the manuscript, mostly related to presentations and discussions of the results.

Major comments and suggestions:

1) Balloon-borne measurements have their strengths and weaknesses. When making interpretation, it is important to recognize the main weakness that the data is approximately one dimensional while the atmosphere in general is described in 3 (spatial) + 1 (time) dimensions. In this specific study, the location of measurements is uniquely situated in the region of steep elevation change. Associated with the terrain variation, the upper level anticyclone also creates a significant tropopause height variation. How the measurement location is relative to the horizontal structure of the tropopause height, especially the region of the highest tropopause, is very important for the conclusions. This consideration is largely missing in the discussion.

Suggestion: Discuss your results in contrast to the results from previously published work using data from balloon-borne measurements with similar payloads but launched from the Tibetan plateau (Bian et al., 2012). Identify the key differences and their implications to your conclusions in relation to the UTLS structure.

2) When concluding the role of ASM in moistening the stratosphere, it is important to recognize that the time scale changes at the level around the CPT. While the vertical transport up to the CPT is in general within the season, it becomes much slower above. The significant difference between the "confined layer" and the "background stratosphere (FLS)", defined to be above the level $\sim$ 65 hPa, is part of the "tape recorder" structure, i.e. the summer and winter difference. How does the ASM enhance this difference is the relevant question.

Suggestions: For the structure of the water vapor tape recorder, a good recent figure

could be the Fig.2 of Glanville and Birner (2017). To estimate how much ASM is more effective in moistening the stratosphere compared to the tropical equatorial entry point in summer, you could possibly use the published result in Bian et al., 2012 (Fig 5) where soundings from Costa Rica (TC4) are used as a contrast to the ASM.

Additional comments:

3) This manuscript is desired to be more concise. For example, it is not clear how section 3 is contributing to the goal of the paper, since the discussion there are not related to particular scientific questions. The points made in that section may be better received when addressing particular questions in the later sections. Also suggest that you work to reduce the repetition of figures.

Suggestions: Be clear on the key objectives of the paper and focus on what serves these objectives. For example, is the comparison of the mean profiles with ECMWF necessary for the objective? Also note that some campaign specifics appeared three times (abstract, intro and campaign description) .

4) When describing the dynamical settings and seasonal changes, it is important to connect to the seasonal change of the ITCZ. See schematic in Lawrence and Lelieveld (2010) and Pan et al 2016 for related discussions. This will put the change from August to November into the right context. It is also more desirable to show the cold point tropopause in ECMWF in addition to PV, since CPT is what you use with the observation.

Specific comments and suggestions:

- References:

P2L13: consider replacing Park 2007 by Hoskins and Rodwell 1995.

P2l23: add Ungermann et al., 2013 before Fadnavis

- Wordings:

P1L22: "It is known to be enriched" -> "It is observed from satellite to contain enhanced"

P1L24: remove "very"

P2L4: reconsider "depletion" – the low ozone is not due to depletion but lofted low ozone air.

P3L11: rephrase "notoriously hardly accessible"

P5L2: rephrase "too high . . ." F. P. temperature and "too high" w.v. mxing ratio. Perhaps "w.v. mr derived by the f. p. t are too high to be physical"?

P14L26-28: A more accurate statement here should be "it is interesting to contrast the result from Pan et al. 2014, where a smaller variability of water vapor is found above the CPT ..."

Comparing the water vapor range of variability in Pan et al. 2014 (figs. 6&7, ∼ 3-5 ppmv) with results from this work (DK17 is ∼ 3.5-6.5 ppm, and NT16 is similar to the Kunming), the variabilities are qualitatively the same. Not sure where you found the "no variability above CPT" as a common concept.

- Figures:

Fig.2: (a) Consider adding easterly jet, which will show where the sounding location was in relation to the anticyclone. (b) Also consider adding simple dynamical field, GPH or tropopause 100 hPa contour to the maps on the right to indicate the anticyclone and possibly the region of highest tropopause. (c) it may be more insightful to color the trajectories by potential temperature.

Figs. 6-7: there is a strong discontinuity between the two figures when you changed from pressure to altitude. Suggest you consider using pressure altitude when you can label the profiles using both pressure and altitude. This can be consistently done throughout the paper.

Fig. 8. There is an error in the caption: FLS should not be CPT to CPT+5 km. By

definition, FLS is above the TOC. I hope this is only an error in description, not in the actual calculation.

- Additional References:

Glanville, A. A. and Birner, T.: Role of vertical and horizontal mixing in the tape recorder signal near the tropical tropopause, Atmos. Chem. Phys., 17, 4337-4353, https://doi.org/10.5194/acp-17-4337-2017, 2017.

Hoskins, B. J., and M. J. Rodwell (1995), A model of the Asian summer monsoon: I—The global scale, J. Atmos. Sci., 52, 1329–1340.

Ungermann, J., Ern, M., Kaufmann, M., Müller, R., Spang, R., Ploeger, F., Vogel, B., and Riese, M.: Observations of PAN and its confinement in the Asian summer monsoon anticyclone in high spatial resolution, Atmos. Chem. Phys., 16, 8389-8403, doi:10.5194/acp-16-8389-2016, 2016.

---

## Referee Comment (RC2) · Anonymous Referee #2 · 17 May 2018

**General comments**

Brunamonti et al. present results from the StratoClim balloon campaigns. They measured vertical distributions of temperature, ozone, water vapour and aerosol in the south Asian UTLS during one post-monsoon and two monsoon campaigns. They identify three significant thermodynamic levels and layers, which provide a framework to understand the UTLS structure within the Asian summer monsoon anticyclone. The paper is sound and clearly within the scope of ACP(D). It is based on a new and important data set that needs to be published.

[Figure]

Some arguments regarding the confinement effect of the ASMA are not yet clear to me, or at least do not sufficiently consider alternative explanations: Convective height might primarily control H2O, O3, and confinement. The effect of confinement on H2O and O3 needs to be clarified. Details are given in the specific comments.
It's hard to tell whether addressing those requires minor or major revisions.
Alternatively, the paper would be worth publishing even without discussing the relative importance of confinement and other processes in the ASMA. The outlook at the end of the paper shows its importance for ongoing other studies.

**Presentation**

(1) Too many acronyms make the paper hard to read. I suggest to count the number of occurrences of each (not well established) acronym, then write out those 50 % that occur least.
(2) Consider to reduce redundancy in the figures (e.g. T vs p for DK17 and NT16AUG is shown in Figs. 1, 4, 6a, 13).
(3) Consider to annotate curves etc. in the figures only, rather than the captions. For instance in Fig.13, the campaigns and the meaning of the colours impede reading of the caption, but are already obvious from the panels.

**Specific comments**

Line numbers in the following are approximate, sometimes referring to the arguments of an entire paragraph.

P3L7: What about aircraft measurements? CARIBIC provides a lot of species in high resolution. Dedicated campaigns (ESMVal/HALO, OMO/HALO, Strato-Clim/Geophysica) sampled higher altitudes and also did some profiles. There are a

few aircraft in-situ monsoon papers, at least from CARIBIC and ESMVal.

P7L2: Fig. 2 shows snapshots of individual days. Are those days chosen to be representative in some respect? Please consider to provide time averages for the respective measurement periods (or for the sounding days).

P7L9: There are different PV thresholds for the dynamical tropopause. Please provide a reference or justify your choice.

P7L17: Given the structural differences of the tropopause region between summer and autumn: Why do you choose the same pressure altitude to compare the two seasons? You might consider to show trajectories started over some altitude range, or from a specific distance to the respective tropopause altitudes.

P8L23: What is the spacing between trajectory starting points?

P8L26: The ASMA box seems to be rather big. Please justify or provide a reference.

P9L19: Formulation for O3 is ambiguous. Please revise.

P10L25: What do you mean by "feature" here: H2O max, O3 min, or the combination of both? Anyway, neither the H2O feature, nor the O3 feature is necessarily related to differences in the strength of convection alone. The time since the last convective influence on the air mass might also be important. If NT16Aug by chance sampled older air on average, the H2O feature would have been smoothed out. Also, convection increases the availability of O3 precursors, leading to enhanced photochemical O3 production. The absence of an O3 minimum just above the LRM in NT16Aug might be due to longer confinement or to higher O3 production. Please discuss.

P11L5: This is consistent to older samples in NT16Aug.

P11L8: This argument is not quite clear to me. H2O in the CLS is compared to H2O in higher altitudes. The difference is attributed to the horizontal confinement effect of the ASMA. However, first order this might just reflect the decreasing frequency of convective tops with altitude. To quantify horizontal (isentropic) confinement, you might consider comparing back-trajectories according to their respective lengths in the ASMA. Please discuss.

P11L14: Not necessarily, see previous comments on convective strength versus age. Age is related to confinement. Please disentangle.

P11L16: Confinement tends to increase O3 via photochemical production. Please discuss horizontal confinement versus the altitude profile of convective influence. The argument regarding the quality of the TOC definition could go the other way round i.e. (simplified): ASMA is driven by convection -> convection reaches to a certain altitude -> no confinement above convective influence.

P11L20. Ditto.

P11L28: Could different temperatures or different ages (time since last convective influence) be alternative explanations for the difference between DK17 and NT16Aug?

P12L19: Comparing NT16Aug to NT16Nov per se generally reflects seasonal variation. Air mass origins might be totally different in August and November, even if there was no ASMA confinement in August. Please reformulate or elaborate, why NT16Aug without ASMA would be like NT16Nov.

P12L29: The parameters affecting the threshold depend on region and season. Is the threshold of Vernier et al. applicable to your measurements without adjustments?

P13L21: Could the thermodynamic conditions at the CPT enhance aerosol formation from gaseous precursors? In that case convective outflow or confinement might not be as important.

P13L22: "its level": Does this refer to confinement or aerosol enhancement? Please reformulate.

P13L25: There is no convective supply of aerosols/precursors in November. Additionally there is no confinement. The ASMA might to some degree enhance ATAL. Consider revising to clarify causes and effects.

P14L33: Please also discuss alternatives to confinement.

P15L3: Please discuss convective height versus confinement.

P15L6: Comparing different altitudes is of limited use for estimating the effects of confinement.

P15L7: Please also consider convective height and possible vertical variations of aerosol formation.

P24, Fig. 2: White contours for water vapour are not discernible. Please revise, e.g. consider omitting or doing an extra plot for them. Insets in panels a, c, e are too small. Consider to include those lines in the right column's panels.

P25, caption of Fig. 2: "Note that in panel c (NT16Nov), trajectories stared ..."
- Panel c is not about trajectories.
- stared -> started
- ppmm -> ppm

P27, Fig. 4: Consider to use ECMWF data only from the times of the respective soundings.

P30, caption of Fig. 6: Consider to give a short explanation of "GPS geometric altitude"

---

## Author Response (AR1)

Below are the comments from the referee in black and replies from the authors in blue

**General Comments**

This manuscript presents the data and analyses of balloon-borne measurements from Northern India and on the southern slope of the Tibetan plateau during two Asian summer monsoon (ASM) seasons. The high vertical resolution profiles of temperature, water vapor, ozone, as well as the cirrus clouds and aerosol information are analyzed together to characterize the region's UTLS thermal and dynamical structure, transport characteristics, in particular the transport of water vapor into the stratosphere and the presence of the Asian tropopause aerosol layer (ATAL). This work is part of a larger project involving the airborne campaign StratoClim. The data and the analyses are well documented in the manuscript. The result contributes important new information to the larger picture of ASM UTLS transport. The work is high quality and fits the scope of ACP well. I have a number of suggestions for improving the manuscript, mostly related to presentations and discussions of the results.

We are grateful to Anonymous Referee #1 for the careful reading and for providing many valuable suggestions, which contribute to improving the manuscript significantly.

We recognize that the points raised by the reviewer about the 3D structure of ASMA are valid, and particularly relevant for assessing the moistening mechanisms of the CLS. This issue is now addressed in the conclusions, with considerations about adiabatic transport from the Tibetan plateau to the southern slopes of the Himalayas and CPT variability based on existing literature. We also recognize that the assessment of the ASM role in moistening the stratosphere is only partly supported by the analysis in this paper, which does not take into account horizontal motion of the air, and consequently this statement was revised in the conclusions.

The abstract was revised in order to better highlight the objectives of this paper, namely to provide an overview of all the balloon measurements performed during our campaigns and to address the broad relevance of this dataset. More targeted studies addressing the question of stratospheric moistening are still ongoing, and the results will be discussed in future publications.

As suggested by the reviewer, the meteorological overview was improved, in particular the seasonal variability section and Figure 2, now showing time-averaged cross sections (including the easterly jet position) and geopotential height fields along with the trajectories. We made the formulation of the proposed UTLS structure more handy by rearranging Sections 4.3-5.2 and introducing the schematics (now Figure 6) earlier in the paper. This includes reducing the use of not well established acronyms (now also summarized in Table 1) and avoiding redundancy in figures (comparison with ECMWF figure removed). The revised manuscript is more concise than the previous version, which was achieved by avoiding repetitions (e.g. campaign specifics) and making the overall discussion more targeted to the objectives of this paper.

In the following, we reply point-by-point to the reviewer's comments, and highlight the corresponding changes made to the manuscript. Note that page and line numbers given in the replies refer to the revised version of the manuscript without tracked-changes.

**Major comments and suggestions:**

**1)** Balloon-borne measurements have their strengths and weaknesses. When making interpretation, it is important to recognize the main weakness that the data is approximately one dimensional while the atmosphere in general is described in 3 (spatial) + 1 (time) dimensions. In this specific study, the location of measurements is uniquely situated in the region of steep elevation change.

Associated with the terrain variation, the upper level anticyclone also creates a significant tropopause height variation. How the measurement location is relative to the horizontal structure of the tropopause height, especially the region of the highest tropopause, is very important for the conclusions. This consideration is largely missing in the discussion.

**Suggestion:** Discuss your results in contrast to the results from previously published work using data from balloon-borne measurements with similar payloads but launched from the Tibetan plateau (Bian et al., 2012). Identify the key differences and their implications to your conclusions in relation to the UTLS structure.

We agree with the reviewer that considering the 3D structure of ASMA, and particularly the "bulging" CPT above the Tibetan plateau, is important for assessing the mechanisms moistening of the CLS. The fact that isentropic transport from Tibetan plateau / below CPT to southern slopes of Himalayas / above CPT might be responsible for (part of) the enhanced H2O observed in the CLS is an important feature that was missing in our previous discussion. This issue is now discussed in the conclusions section, based on comparison of the average CPT isentropic levels from our datasets with Tibetan plateau soundings from Bian et al. (2012) as well as with considerations about CPT variability based on previous literature (page 14, lines 3-9).

Comparison of our southern-slopes measurements with the simultaneous Lhasa 2016 and Kunming 2017 campaigns of SWOP in the Tibetan plateau region are ongoing (see page 15 lines 1-2) and the results, which will address explicitly the issue of adiabatic transport vs slow ascent and overshooting convection (i.e. "chimney vs blower"), will be discussed in a future publication.

**2)** When concluding the role of ASM in moistening the stratosphere, it is important to recognize that the time scale changes at the level around the CPT. While the vertical transport up to the CPT is in general within the season, it becomes much slower above. The significant difference between the "confined layer" and the "background stratosphere (FLS)", defined to be above the level ~ 65 hPa, is part of the "tape recorder" structure, i.e. the summer and winter difference. How does the ASM enhance this difference is the relevant question.

**Suggestions:** For the structure of the water vapor tape recorder, a good recent figure could be the Fig.2 of Glanville and Birner (2017). To estimate how much ASM is more effective in moistening the stratosphere compared to the tropical equatorial entry point in summer, you could possibly use the published result in Bian et al., 2012 (Fig 5) where soundings from Costa Rica (TC4) are used as a contrast to the ASM.

We agree with the reviewer that, for assessing the role of ASMA in moistening the stratosphere, comparing different stacks of altitudes is of limited use due to the tape recorder structure, and that the fate of the enhanced  $H_2O$  observed in the CLS needs to be addressed by explicitely taking into account the horizontal motion of the air.

Consequently, the comparison of water vapor PDFs in the CLS and free stratosphere region was removed from the manuscript, and replaced with a discussion based on results from recent literature (namely, Pan et al., 2016) (page 14 lines 10-17). In this paragraph, the statement of ASMA moistening the stratosphere is flagged as "potential" (page 14 line 10).

As mentioned in the manuscript (page 2 lines 5-6 and page 14 lines 32-33), further investigations aimed to assess the relevance of our measurements in the context of stratospheric moistening and related processes are currently ongoing. However, we refrain from performing additional analyses here, as we intend to discuss this issue in a future dedicated publication.

**Additional comments:**

**3)** This manuscript is desired to be more concise. For example, it is not clear how section 3 is contributing to the goal of the paper, since the discussion there are not related to particular scientific questions. The points made in that section may be better received when addressing particular questions in the later sections. Also suggest that you work to reduce the repetition of figures.

**Suggestions:** Be clear on the key objectives of the paper and focus on what serves these objectives. For example, is the comparison of the mean profiles with ECMWF necessary for the objective? Also note that some campaign specifics appeared three times (abstract, intro and campaign description).

Several minor and major changes were made in order to make the manuscript more concise and more targeted to its objectives, and to avoid redundancy in figures. In particular:

- Abstract simplified (use of acronyms reduced)
- Figure 4 (comparison with ECMWF) removed
- Figure 6 reduced (panel a eliminated, panel b given as Figure S4 in supplementary material)
- Use of acronyms reduced (FLS eliminated, use of UT and LS as individual acronyms avoided)
- Main acronyms summarized in Table 1
- Sections 4.3-5.2 rearranged and schematics figure introduced earlier in the paper (Figure 6)
   Campaign specifics removed from abstract and introduction

As a consequence, the revised manuscript is shorter than the previous version (despite several other additions requested by the reviewers were made), and we believe the presentation of the UTLS structure that we define is more fluent and reader-friendly.

**4)** When describing the dynamical settings and seasonal changes, it is important to connect to the seasonal change of the ITCZ. See schematic in Lawrence and Lelieveld (2010) and Pan et al 2016 for related discussions. This will put the change from August to November into the right context. It is also more desirable to show the cold point tropopause in ECMWF in addition to PV, since CPT is what you use with the observation.

According to this comment, the discussion of dynamical settings and seasonal variability was improved with references to Lawrence and Lelieveld (2010) and Munchak and Pan (2014), relating the observed features to the seasonal variations of the ITCZ and the jet streams (page 5 lines 22-23, page 5 lines 30-32 and page 7 lines 14-16). In addition, a sketch of the ITCZ is now shown in schematics of the UTLS structure (Figure 6). Figure 2 was also improved to facilitate the discussion of the seasonal variability (see details in specific comment below).

**Specific comments and suggestions:**

**- References:**

P2L13: consider replacing Park 2007 by Hoskins and Rodwell 1995.

Done (page 2 line 10).

**P2I23: add Ungermann et al., 2013 before Fadnavis**

Done (however, note that it is Ungerman et al., 2016, added after Fadnavis et al., 2013 to maintain chronological order) (page 2 line 19).

**- Wordings:**

P1L22: "It is known to be enriched" -> "It is observed from satellite to contain enhanced"

Done ("be enriched" -> "contain enhanced") (page 1 line 22).

P1L24: remove "very"

Done (page 1 line 25).

P2L4: reconsider "depletion" – the low ozone is not due to depletion but lofted low ozone air.

The whole sentence was removed in new abstract.

P3L11: rephrase "notoriously hardly accessible"

Done (page 3 line 9).

P5L2: rephrase "too high" F. P. temperature and "too high" w.v. mixing ratio. Perhaps "w.v. mr derived by the f. p. t. are too high to be physical"?

Done (page 4 lines 32-33).

P14L26-28: A more accurate statement here should be "it is interesting to contrast the result from Pan et al. 2014, where a smaller variability of water vapor is found above the CPT ..." Comparing the water vapor range of variability in Pan et al. 2014 (figs. 6&7, \_ 3-5 ppmv) with results from this work (DK17 is \_ 3.5-6.5 ppm, and NT16 is similar to the Kunming), the variabilities are qualitatively the same. Not sure where you found the "no variability above CPT" as a common concept.

The whole sentence was removed in new conclusions section.

**- Figures:**

Fig.2: (a) Consider adding easterly jet, which will show where the sounding location was in relation to the anticyclone. (b) Also consider adding simple dynamical field, GPH or tropopause 100 hPa contour to the maps on the right to indicate the anticyclone and possibly the region of highest tropopause. (c) it may be more insightful to color the trajectories by potential temperature.

Figure 2 was subject to a number of improvements accordingly:

- Wind contours at 20 m/s and 30 m/s added, showing the easterly jet position (panels a-c-e)

- X-axis range enlarged to show latitudes 10-50°N (panels a-c-e)

- Individual days replaced with time-average of the respective campaign periods (panels a-c-e)

- Geopotential height at 100 hPa (time average of each measurement period) added to the trajectory panels, showing the region of highest tropopause (panels b-d-f)

As a consequence, Figure 2 now provides a significantly higher amount of information than the previous version, and the discussion of seasonal variability was improved accordingly (page 6 line 26 to page 7 line 6).

Figs. 6-7: there is a strong discontinuity between the two figures when you changed from pressure to altitude. Suggest you consider using pressure altitude when you can label the profiles using both pressure and altitude. This can be consistently done throughout the paper.

Unfortunately we do not understand this comment properly, and in particular it is not clear to us what the reviewer means by "discontinuity". Figure 7 shows altitude relative to CPT as y-axis (not just altitude), therefore adding an altitude scale to all plots would not make the profiles look more similar. In addition, all figures in the paper show more than one dataset at the same time, hence adding an altitude axis along with the pressure profiles is strictly speaking not possible. Therefore, we refrain from applying any changes.

Fig. 8. There is an error in the caption: FLS should not be CPT to CPT+5 km. By definition, FLS is above the TOC. I hope this is only an error in description, not in the actual calculation.

The reviewer is right and the description was corrected accordingly (the actual calculation was correct). Note that the definitions of the free stratosphere and troposphere regions for the PDFs calculation are now given in Section 5.3 (page 10 lines 2-4) instead of in the caption of Figure 8.

**Anonymous Referee #2**

Below are the comments from the referee in black and replies from the authors in blue

**General comments**

Brunamonti et al. present results from the StratoClim balloon campaigns. They measured vertical distributions of temperature, ozone, water vapour and aerosol in the south Asian UTLS during one post-monsoon and two monsoon campaigns. They identify three significant thermodynamic levels and layers, which provide a framework to understand the UTLS structure within the Asian summer monsoon anticyclone. The paper is sound and clearly within the scope of ACP(D). It is based on a new and important data set that needs to be published.

Some arguments regarding the confinement effect of the ASMA are not yet clear to me, or at least do not sufficiently consider alternative explanations: Convective height might primarily control H2O, O3, and confinement. The effect of confinement on H2O and O3 needs to be clarified. Details are given in the specific comments.

It's hard to tell whether addressing those requires minor or major revisions. Alternatively, the paper would be worth publishing even without discussing the relative importance of confinement and other processes in the ASMA. The outlook at the end of the paper shows its importance for ongoing other studies.

We are grateful to Anonymous Referee #2 for the careful reading and for providing many valuable suggestions, which contribute to improving the manuscript significantly.

We recognize that the points raised by the reviewer concerning the relative importance of confinement and convective altitude are valid, as well as the considerations made in specific comments regarding age of air and convection. We improved the discussion of the  $H_2O$ ,  $O_3$  and aerosol vertical distributions by taking into account the suggested alternative explanations (mainly Sections 5.3 and 6.2), and our conclusions were revised accordingly.

As suggested by the reviewer, the use of acronyms was reduced for easier reading (including the abstract, which was substantially simplified) and redundancy in figures was avoided. A new table was introduced to summarize the main not well established acronyms (Table 1). The formulation of the UTLS structure that we propose was made more fluent by rearranging Sections 4.3-5.2 and by introducing the schematics figure (now Figure 6) earlier in the paper.

The meteorological overview section was also improved, and particularly Figure 2, now displaying time-averaged cross sections and geopotential height fields along with the trajectories.

Despite several additions (requested by the reviewers) were made, the revised manuscript is more concise than the previous version. This was achieved by avoiding repetitions (e.g. the campaign details) and making the overall discussion more targeted to the objectives of this paper.

In the following, we reply point-by-point to the reviewer's comments, and highlight the corresponding changes made to the manuscript. Note that page and line numbers given in the replies refer to the revised version of the manuscript without tracked-changes.

**Presentation**

(1) Too many acronyms make the paper hard to read. I suggest to count the number of occurrences of each (not well established) acronym, then write out those 50 % that occur least.

The use of acronyms was reduced in the manuscript accordingly, namely by eliminating FLS (free lower stratosphere) and avoiding the use of UT and LS as individual acronyms (e.g. CLS is now used consistently throughout the whole paper, avoiding the use of "Confined LS"). In addition, a new table was introduced to summarize the main not-well established acronyms used in the paper

(Table 1), and the schematics in former Figure 13 was introduced earlier in the paper (now Figure 6) to help the reader familiarize with the acronyms.

In order to make the manuscript more reader-friendly and the formulation of the UTLS structure that we propose more fluent, we have also moved former section 4.3 ("Horizontal confinement in ASMA") to section 5.3 ("Confined lower stratosphere"), where the acronyms TOC and CLS are defined and the schematics in Figure 6 is discussed.

Finally, the abstract was also simplified by strongly reducing the use of acronyms in it.

(2) Consider to reduce redundancy in the figures (e.g. T vs p for DK17 and NT16AUG is shown in Figs. 1, 4, 6a, 13).

Redundancy in figures was reduced accordingly:

- Figure 4 (comparison with ECMWF) was removed, as it was not directly necessary to the objectives of the paper

- Figure 6 reduced: panel a (T vs p) was removed due to redundancy, and panel b ( $\theta$  vs p) moved to supplementary material (Figure S4).

(3) Consider to annotate curves etc. in the figures only, rather than the captions. For instance in Fig.13, the campaigns and the meaning of the colours impede reading of the caption, but are already obvious from the panels.

In Figure 13 the curves are identified by the color of their axis label, and since the figure is already quite "full", we believe there is no need to add an extra legend to it (we tried different options but figure becomes too hard to read). In the revised version, Figure 13 was improved with respect to the previous version ( $\theta$  dashed lines changed from green to grey, labels rearranged).

**Specific comments**

Line numbers in the following are approximate, sometimes referring to the arguments of an entire paragraph.

P3L7: What about aircraft measurements? CARIBIC provides a lot of species in high resolution. Dedicated campaigns (ESMVal/HALO, OMO/HALO, StratoClim/Geophysica) sampled higher altitudes and also did some profiles. There are a few aircraft in-situ monsoon papers, at least from CARIBIC and ESMVal.

Reference to aircraft measurements in ASMA, namely from ESMVal (Gottschald et al., 2018) and CARIBIC (Raute-Schöch et al., 2016) added to the introduction accordingly (page 3 lines 9-11).

P7L2: Fig. 2 shows snapshots of individual days. Are those days chosen to be representative in some respect? Please consider to provide time averages for the respective measurement periods (or for the sounding days).

Change made accordingly: individual days replaced by time average of each measurement period (Figure 2, panels a, c, e).

P7L9: There are different PV thresholds for the dynamical tropopause. Please provide a reference or justify your choice.

Reference to Kunz et al. (2011) for the choice of the PV threshold for the dynamical tropopause was added, and the discussion was revised accordingly: we now use PV = 3-4 PVU which is more appropriate for the considered latitude and season (page 7, lines 5-6).

P7L17: Given the structural differences of the tropopause region between summer and autumn: Why do you choose the same pressure altitude to compare the two seasons? You might consider to show trajectories started over some altitude range, or from a specific distance to the respective tropopause altitudes.

Figure 2 is meant to illustrate the differences between the UTLS structure and dynamical features of the monsoon vs. post-monsoon season in a qualitative manner. The contrast between monsoon and post-monsoon season is already evident by comparing panels b-f vs. panel d (furthermore now that geopotential height fields are also shown in addition to trajectories). Therefore, we believe that there is no need to further refine the trajectory comparison.

P8L23: What is the spacing between trajectory starting points?

Trajectories are initialized at 5 hPa intervals. Manuscript modified to include this information (page 9 line 5).

P8L26: The ASMA box seems to be rather big. Please justify or provide a reference.

Our ASMA box definition (10-50°N, 0-140°E) is based on the average geopotential height fields during our campaign periods, now shown in Figure 2 (page 9 line 11 rephrased accordingly). Similarly large domains were used to approximate the ASMA area in previous literature, e.g.:

- 15-45°N, 5-105°E (Vernier et al., 2011)

- 10-60°N, 10-160°E (Ploeger et al., 2015)

- 0-60°N, 0-140°E (Pan et al., 2016)

Therefore we believe our boundaries provide a reasonable approximation of the ASMA area.

P9L19: Formulation for O3 is ambiguous. Please revise.

The whole sentence was removed from the revised version (for the sake of brevity).

P10L25: What do you mean by "feature" here: H2O max, O3 min, or the combination of both? Anyway, neither the H2O feature, nor the O3 feature is necessarily related to differences in the strength of convection alone. The time since the last convective influence on the air mass might also be important. If NT16Aug by chance sampled older air on average, the H2O feature would have been smoothed out. Also, convection increases the availability of O3 precursors, leading to enhanced photochemical O3 production. The absence of an O3 minimum just above the LRM in NT16Aug might be due to longer confinement or to higher O3 production. Please discuss.

We agree with the reviewer that considering the age of air (meant as time elapsed since the last convective influence) is important due to enhanced photochemical  $O_3$  production in ASMA, and that the fact of missing  $O_3$  minimum above the LRM in the mean profile of NT16AUG is consistent with older air sampled on average during this campaign vs. fresh convective outflow sampled more frequently during DK17. We also recognize that the same argument applies to the H2O maximum above the CPT (see comment below).

Assuming the balloon soundings are frequent enough to be statistically representative of the respective measurement periods, we still would argue that this evidence does suggests that more frequent deep convection occurred during DK17 compared to  $NT16_{AUG}$ .

The manuscript was revised according to this consideration (page 10 lines 21-26).

P11L5: This is consistent to older samples in NT16Aug.

Same arguments as in the comment above applies to the H2O maximum above the CPT in DK17: we agree with the reviewer and the manuscript was modified accordingly (page 10 lines 24-25).

P11L8: This argument is not quite clear to me. H2O in the CLS is compared to H2O in higher altitudes. The difference is attributed to the horizontal confinement effect of the ASMA. However, first order this might just reflect the decreasing frequency of convective tops with altitude. To quantify horizontal (isentropic) confinement, you might consider comparing back-trajectories according to their respective lengths in the ASMA. Please discuss.

We agree with the reviewer that, to some extent, decreasing "convective signature" with altitude might simply reflect the decreasing frequency of overshooting convective tops with altitude. Indeed, the current status of our analysis does not explicitly disentangles the effects of convective vs. confinement top height in controlling the vertical distributions of  $H_2O$ ,  $O_3$  and aerosols above the CPT. Furthermore, the anticyclonic confinement is caused by the high pressure built up by the deep convection, hence the two processes are intrinsecally connected.

However we also consider that, to our knowledge, convective updrafts overshooting the CPT by 1.5-2 km were never observed (also not by the Geopyhsica campaign during StratoClim in Nepal), and that the gradients in the vertical distributions of  $H_2O$ ,  $O_3$  and aerosols are in good agreement with the TOC inferred from backward trajectories.

Therefore, we still would argue that these evidences suggest that the horizontal confinement effect of ASMA plays an important role in shaping the vertical distributions of  $H_2O$  and  $O_3$  above the CPT (and in particular the  $H_2O$  enhancement in the CLS). Nevertheless, we recognize that further analysis would be required to fully disentangle the relevance of the different transport processed, and we revised the manuscript accordingly (page 10 line 32 to page 11 line 5).

P11L14: Not necessarily, see previous comments on convective strength versus age. Age is related to confinement. Please disentangle.

Same as above: we agree with the reviewer and the manuscript was modified accordingly (page 10 lines 25-26).

P11L16: Confinement tends to increase O3 via photochemical production. Please discuss horizontal confinement versus the altitude profile of convective influence. The argument regarding the quality of the TOC definition could go the other way round i.e. (simplified): ASMA is driven by convection -> convection reaches to a certain altitude -> no confinement above convective influence.

Same as above: we agree (at least to some extent) with the reviewer, and the manuscript was modified accordingly (page 11 lines 3-5).

**P11L20. Ditto.**

Same as above: the statement that TOC controls the vertical distributions of  $H_2O$  and  $O_3$  was removed and replaced with a more detailed discussion (page 10 line 32 to page 11 line 5).

P11L28: Could different temperatures or different ages (time since last convective influence) be alternative explanations for the difference between DK17 and NT16Aug?

We believe that different temperatures are the main driver of the different ice saturations measured during  $NT16_{AUG}$  and DK17, as it is stated already in the manuscript (page 11 line 9). At the current status of our analysis, it is not obvious to infer a correlation between of age of air and ice saturation. The statement that higher ice saturations in DK17 are related to stronger convective activity was removed from the manuscript.

P12L19: Comparing NT16Aug to NT16Nov per se generally reflects seasonal variation. Air mass origins might be totally different in August and November, even if there was no ASMA confinement in August. Please reformulate or elaborate, why NT16Aug without ASMA would be like NT16Nov.

We fully agree with the reviewer that, in addition to confinement (or lack of thereof), air mass origin and potential direct exposure to deep convection is important, as it provides the supply of aerosols and precursor gases to the UTLS. The manuscript was revised accordingly (page 12 lines 1-3).

P12L29: The parameters affecting the threshold depend on region and season. Is the threshold of Vernier et al. applicable to your measurements without adjustments?

The parameters affecting the cloud-filtering thresholds are independent of region and season. These thresholds are just based on optical considerations and the typical size ranges of atmospheric aerosols and ice crystals (see page 12 lines 10-12). Furthermore, it happens that the thresholds by Vernier et al. (2015) have been developed based on measurements from the same region and season as our measurements (Lhasa, China during the ASM season), and comply with CALIPSO depolarization criteria. Therefore, we are confident that these thresholds are applicable to our measurements without adjustments.

P13L21: Could the thermodynamic conditions at the CPT enhance aerosol formation from gaseous precursors? In that case convective outflow or confinement might not be as important.

We agree with the reviewer that thermodynamic conditions (namely colder temperatures) at the CPT can enhance the partitioning of condensable material (e.g. nitrate) to the aerosol phase, and therefore the fact that ATAL shows maximum BSR at the CPT is likely also influenced by temperature, in addition to confinement. The manuscript was revised accordingly (page 13 lines 5-7).

P13L22: "its level": Does this refer to confinement or aerosol enhancement? Please reformulate.

Maximum strength refers to confinement. Sentence rephrased accordingly (page 13 lines 5-7).

P13L25: There is no convective supply of aerosols/precursors in November. Additionally there is no confinement. The ASMA might to some degree enhance ATAL. Consider revising to clarify causes and effects.

Same as discussed above: we agree with the reviewer that supply/lack of aerosols and precursors via deep convection needs to be considered in addition to confinement. Sentence revised accordingly (page 13 line 12).

P14L33: Please also discuss alternatives to confinement.

Alternatives to confinement (supply/lack of aerosols and precursors via deep convection) already discussed in Section 6.2 (see comments above). Conclusions revised accordingly (page 14 lines 18-20).

P15L3: Please discuss convective height versus confinement.

Convective height vs. confinement already discussed in Section 5.3 (see comments above). Conclusions revised accordingly (page 13 line 30 to page 14 line 2).

P15L6: Comparing different altitudes is of limited use for estimating the effects of confinement.

As discussed above, we agree with the reviewer that comparing different altitudes is of limited use for estimating the effects of confinement. This statement was removed from the conclusions.

P15L7: Please also consider convective height and possible vertical variations of aerosol formation.

Same as above: alternatives to confinement (convective height, supply/lack of aerosols and precursors via deep convection) now discussed throughout Sections 5.3 and 6.2. This statement was removed from the conclusions.

P24, Fig. 2: White contours for water vapour are not discernible. Please revise, e.g. consider omitting or doing an extra plot for them. Insets in panels a, c, e are too small. Consider to include those lines in the right column's panels.

White contours for IWC eliminated in the revised version of Figure 2.

P25, caption of Fig. 2: "Note that in panel c (NT16Nov), trajectories started : : :"

- Panel c is not about trajectories.
- stared -> started
- ppmm -> ppm

Caption of Figure 2 revised and typos corrected.

P27, Fig. 4: Consider to use ECMWF data only from the times of the respective soundings.

Figure 4 was removed for the sake of reducing redundancy in figures.

P30, caption of Fig. 6: Consider to give a short explanation of "GPS geometric altitude"

Former Figure 6 (now Figure 4) was revised and the caption rephrased to avoid mentioning "GPS" geometric altitude, but just "geometric altitude" as it is done previously in the main text (see page 5 line 12).

**Balloon-borne measurements of temperature, water vapor, ozone and aerosol backscatter at the southern slopes of the Himalayas during StratoClim 2016-2017**

- 5 Simone Brunamonti1, Teresa Jorge1, Peter Oelsner2, Sreeharsha Hanumanthu3,4, Bhupendra B. Singh3, K. Ravi Kumar3,10, Sunil Sonbawne3, Susanne Meier2, Deepak Singh5, Frank G. Wienhold1, Bei Ping Luo1, Maxi Boettcher1, Yann Poltera1, Hannu Jauhiainen8, Rijan Kayastha6, Jagadish Karmacharya9, Ruud Dirksen2, Manish Naja5, Markus Rex7, Suvarna Fadnavis3 and Thomas Peter1
- 1Institute for Atmospheric and Climate Science (IAC), Swiss Federal Institute of Technology (ETH), Zürich, Switzerland
   2Deutscher Wetterdienst (DWD) / GCOS Reference Upper Air Network (GRUAN) Lead Center, Lindenberg, Germany
   3Indian Institute of Tropical Meteorology (IITM), Pune, India
   4Forschungszentrum Jülich (FZJ), Institute of the Energy and Climate Research Stratosphere (IEK-7), Jülich, Germany
   5Aryabhatta Research Institute of Observational Sciences (ARIES), Nainital, India
   6Kathmandu University (KU), Dhulikhel, Nepal
- 7Alfred Wegener Institute (AWI) for Polar and Marine Research, Potsdam, Germany
   8Vaisala Oyj, Vantaa, Finland
   9Department of Hydrology and Meteorology (DHM), Kathmandu, Nepal
   10Now at Centre for Atmospheric Sciences, Indian Institute of Technology (IIT), Delhi, India,
- 20 Correspondence to: Simone Brunamonti (simone.brunamonti@env.ethz.ch)

Abstract. The Asian summer monsoon anticyclone (ASMA) is a major meteorological system of the upper troposphere-lower stratosphere (UTLS) during boreal summer. It is known to contain enhanced in tropospheric trace gases and aerosols, due to rapid lifting from the boundary layer by deep convection and subsequent horizontal confinement. Given its dynamical struc-

- 30 shorter post-monsoon campaign in Nainital in November 2016 (NT16NOV). These measurements provide unprecedented insights into the JJTLS thermal structure, the vertical distributions of water vapor, ozone and aerosols, cirrus cloud properties and interannual variability in ASMA. Here we provide an overview of all the data collected during the three campaign periods, with focus on the UTLS region and the monsoon season. We analyze the vertical structure of ASMA in terms of significant levels and layers, identified from the temperature and potential temperature lapse rates and Lagrangian backward trajectories.
- 35 providing a framework for relating the measurements to local thermodynamic properties and the large-scale anticyclonic flow, Both the monsoon-season campaigns show evidence of deep convection and confinement extending up to 1.5-2 km above the

[revised manuscript text omitted]

| 1   | Moved (insertion) [1]                                                                                                                                                                                             |  |  |  |  |  |  |  |
|-----|-------------------------------------------------------------------------------------------------------------------------------------------------------------------------------------------------------------------|--|--|--|--|--|--|--|
| -   | Deleted: 1                                                                                                                                                                                                        |  |  |  |  |  |  |  |
| /   | Deleted: Finally,                                                                                                                                                                                                 |  |  |  |  |  |  |  |
|     | Deleted: n                                                                                                                                                                                                        |  |  |  |  |  |  |  |
|     | Commented [BS1]: Subsections 3.1, 3.2, 3.3 eliminated                                                                                                                                                             |  |  |  |  |  |  |  |
| Δ   | Deleted: General                                                                                                                                                                                                  |  |  |  |  |  |  |  |
|     | Deleted: of the data                                                                                                                                                                                              |  |  |  |  |  |  |  |
|     | Deleted: Note that for improved recognition, we keep this color coding also in subsequent figures.                                                                                                         |  |  |  |  |  |  |  |
| / / | Deleted: 3.1 Troposphere ¶                                                                                                                                                                                        |  |  |  |  |  |  |  |
| //  | Deleted: , seasonal variability of water vapor mixing ratio is the most evident feature, with differences                                                                                                  |  |  |  |  |  |  |  |
|     | Deleted: (close to the moist adiabat)                                                                                                                                                                             |  |  |  |  |  |  |  |
| /   | Deleted: (close to the dry adiabat)                                                                                                                                                                               |  |  |  |  |  |  |  |
|     | Deleted: Additionally, in the upper troposphere, the same feature is likely also related to stronger convective activity occurring during DK17 compared to NT16 AUG (discussed in Section 5.2). |  |  |  |  |  |  |  |
|     | Deleted: 3.2 Tropopause region¶                                                                                                                                                                                   |  |  |  |  |  |  |  |
|     | Deleted: typical of the tropics, the tropopause structure in                                                                                                                                               |  |  |  |  |  |  |  |
|     | Deleted: has more similarity to that in the mid-latitudes, with                                                                                                                                            |  |  |  |  |  |  |  |
| //  | Deleted: )                                                                                                                                                                                                        |  |  |  |  |  |  |  |
| 1   | Deleted: (                                                                                                                                                                                                        |  |  |  |  |  |  |  |

[revised manuscript text omitted]

**Commented [BS3]: Section 4.3 moved to Section 5.2**

**Moved (insertion) [3]**

**Deleted:** Figure 7 highlights the levels and layers used throughout this work (see also the summary in Figure 12 at the end of this paper). Based on the definitions of LRM, CPT and TOC (introduced in Section 4.3), hereafter we will refer to: UT as the region of altitudes below the LRM, Asian TTL (ATTL) as the region between LRM and CPT, Confined L{...

| Deleted: Horizontal confinement in ASMA                               |
|-----------------------------------------------------------------------|
| Deleted: In conclusion of the meteorological overview, w       |
| Deleted: calculate                                                    |
| Deleted: with LAGRANTO using ERA-Interim re-analys                    |
| Deleted: , Panels                                                     |
| Deleted: then                                                         |
| Deleted: T                                                            |
| Deleted: he ASMA area is                                              |
| Deleted: d                                                            |
| Deleted: shown by grey                                                |
| Deleted: lines                                                        |
| Deleted: , Panels b, f                                                |
| Deleted: T                                                            |
| Deleted: are plotted                                                  |
| Deleted: Therefore,                                                   |
| Deleted: the anticyclonic                                             |
| Deleted: of                                                           |
| Deleted: the                                                          |
| Deleted: M                                                            |
| Deleted: 2                                                            |
| Deleted: In Sections 5 and 6, we will show that the TOC        |
| Deleted: 2                                                            |
| Deleted: the vertical distributions and variability of tracers |

of temperature,  $H_2O$  mixing ratio and  $O_3$  mixing ratio in this coordinate system (note that, besides the CPT in black, the average LRM and TOC levels are shown by blue dashed for NT16AUG, and red dashed lines for DK17). Figure 8 shows probability density functions (PDFs) of temperature (left column),  $H_2O$  mixing ratio (center) and  $O_3$  mixing ratio (right column) calculated in the free stratosphere (panels a-c), CLS (d-f), ATTL (g-i) and jroposphere (j-l) regions, for NT16AUG and DK17. The PDFs

of the free stratosphere region are calculated for altitudes between TOC and CPT + 5 km, while the troposphere PDFs between
 CPT - 6 km and LRM, i.e. covering the whole range of altitudes (with respect to CPT) as shown in Figure 7.
 Water vapor mixing ratio in DK17 shows a minimum at the CPT and a local maximum in the CLS (Figure 7b), centered about

1 km above the local CPT (i.e. not the average CPT, but evaluated for each profile individually). The H2O minimum is conceivably due to unusually high frequency of occurrence of ice clouds near the CPT in DK17, depleting water vapor from the /

- 10 gas phase in favour of the condensed phase, hence resulting in a strongly dehydrated CPT (see Section 5.4). The isolated H2O maximum in the CLS is consistent with hydration by overshooting convective updrafts, injecting ice crystals above the CPT, / which then evaporate and and thus release localized "pockets" of moist air, Convective updrafts overshooting the CPT were / observed by Corti et al. (2008), and a similar hydration mechanism was hypothesized by Dauhut et al. (2015; 2016), In both NT16AUG and DK17, the PDFs of H2O mixing ratio show high water vapor in the CLS (Figure 8e) compared to the
- 15 free stratosphere (Figure 8b), In particular, the PDFs in the CLS are broad (3-7 ppmv) and skewed towards high values, while tht, PDFs in the FLS are narrow (3-5 ppmv) and show the expected distribution of background stratospheric H2O mixing ratios, The high H2O mixing ratios in the CLS compared in DK17 are obviously related to the previously discussed isolated maximum, yet enhanced frequency of occurrence of high H2O mixing ratios is also observed in NT16AUG despite no local maximum was found in this dataset. This is consistent with the slow ascent of moist convective outflow air within the confined anticyclone,
- 20 and in part may reflect the decreasing frequency of overshooting convective tops with altitude in NT16AUG.
  Ozone mixing ratio in DK17 shows a minimum slightly above the LRM (Figure 7c), which is characteristic of deep convection, rapidly transporting ozone-poor air from the boundary layer to the convective outflow level (Gettelman and de F. Forster, 2002; Vömel et al. 2002; Paulik and Birner, 2012), suggests that the average age of air, meant as the time elapsed since the last interaction with deep convection, was higher in NT16AUG compared to DK17, such that the O3 minimum is smeared out
- 25 by mixing and additional photochemical production (which is ehnanced in ASMA due to the enrichment in ozone precursors: Gottschaldt et al., 2018). Higher dilution of the convective signature in NT16AUG vs. DK17 is also consistent with the absence on an H2O maximum above the CPT in NT16AUG (Figure 7b), and with the high frequency of occurrence of low O3 mixing ratios in DK17 compared to NT16AUG 
[revised manuscript text omitted]

In both NT16AUG and DK17, high H2O and low O3 were found in the ATTL and CLS, which is the signature of deep convection, extending up to 1.5-2 km above the CPT. Convective features are more pronounced in DK17 compared to NT16AUG, suggesting that convective activity at the southern slopes of the Himalayas was more intense during the ASM season 2017 compared to 2016. In particular, an isolated H2O maximum in the CLS was observed in DK17, which we argue it may be due to overshooting

- 5 convection, as previously observed by Corti et al. (2008) and modelled by Dauhut et al. (2015; 2016). The fact that the average CPTs in our datasets occur at lower potential temperatures than previously found above the Tibetan plateau suggests that, in addition to slow ascent and overshooting convection (discussed in Section 5.3), isentropic transport from the Tibetan plateau (below CPT) to the southern slopes of the Himalayas (above CPT) might also contribute to the high H2O observed in the CLS. Nevertheless, since the isentropic level of the CPT is subject to strong instantaneous perturbations
- associated with convection and wave activity (e.g. Boehm and Verlinde, 2000; Sherwood et al., 2003; Munchak and Pan, 2014; Muhsin et al., 2018), a conclusion based on just the average CPT from a limited number of profiles is to be taken with caution, and further investigations will be required to assess the relevance of the different transport pathways. The high H2O observed in the CLS is particularly interesting because of its potential implications for stratospheric moistening. The air masses in the CLS have already crossed the CPT, hence will be unlikely subject to further dehydration, so it appears
- 15 that the high H2O in this layer is prone to be lifted further and mixed into the (drier) free stratosphere. However, it was shown that vertical transport above the ASMA might not be very efficient due to the slow ascending velocities of the Brewer-Dobson circulation in this region and season, and that the most efficient transport pathway is quasi-horizontal transport through the horizontal boundaries of ASMA and subsequent upwelling in the stratosphere above the deep tropics (Pan et al., 2016). Therefore, the fate of the air masses in the CLS (hence the moistening potential of the high H2O in this layer) needs to be addressed
- 20 by explicitly taking into account the horizontal motion of the air, which we do not investigate in this work. Cloud-filtering of the aerosol backscatter reveals the signature of ATAL, extending from the LRM to the TOC with maximum backscatter at the CPT, and with similar BSR enhancement as in previous measurements from Lhasa, China (Vernier et al., 2015). No aerosol enhancement was found in NT16NOV. In both NT16AUG and DK17, ice saturation is minimum at the LRM and increases in the ATTL, similarly to as in the tropics (Vömel et al., 2002). Due to the much cold temperatures, average Sice
- 25 in DK17 is remarkably higher than in NT16AUG, as well as than in previous measurements from the Tibetan plateau (Bian et al., 2012), Numerous thin cirrus clouds were detected during the NT16AUG and DK17campaigns (often embedded in ATAL), and their optical properties suggest they might be formed by heterogeneous freezing. Our analysis provides a comprehensive and high-resolution overview of the UTLS structure and composition at the southern

slopes of the Himalayas. The thermodynamically-significant levels and layers that we identify offer physically-meaningful

30 boundaries for the interpretation of the observed vertical distributions of water vapor, ozone and aerosols in ASMA, and the extents of enhanced H2O and aerosols (ATAL) above the CPT are in good agreement with the top of anticyclonic confinement estimated from air mass backward trajectories. Our approach based on significant Jevels, rather than fixed pressure or altitude stacks, also provides useful diagnostics for the comparison of our in-situ measurements with global climate model outputs.

**Moved down [6]:** Due to extremely cold temperatures, average  $S_{icc}$  in DK17 is remarkably higher than in NT16AUG, 
[revised manuscript text omitted]
(^{\circ}C)$ | z (km)              | p (hPa)      | $\theta\left(\mathbf{K}\right)$ | $T(^{\circ}C)$ | z (km) | p (hPa) | $\theta\left(\mathbf{K}\right)$ | $T(^{\circ}C)$ |  |
| LRM | 13.3                | 180     | 360          | -52.7          | 10.5                | 260          | 337.5                           | -43.6          | 13.7   | 169.5   | 362.5                           | -55            |  |
| LRT | 17.0                | 97.5    | 382          | -76.8          | 16.0                | 108          | 378                             | -73.2          | 17.6   | 88      | 383.5                           | -81.7          |  |
| СРТ | 17.0                | 97.5    | 382          | -76.8          | 18.5                | 69.5         | 424                             | -75.3          | 17.6   | 88      | 383.5                           | -81.7          |  |
| тос | 18.6                | 73      | 421.5        | -73.7          | N.A.                | N.A . | N.A .                    | N.A.    | 19.5   | 63.5    | 441                             | -72.7          |  |

**Table 3.** Mean values of altitude (z), pressure (p), potential temperature ( $\theta$ ) and temperature (T) of the lapse rate minimum (LRM), lapse rate tropopause (LRT), cold-point tropopause (CPT) and top of confinement (TOC) levels during the three campaign periods, NT16AUG, NT16NOV and DK17. Note that for NT16NOV, the definition of TOC is not applicable (N.A.).

Commented [BS6]: Former Table 2 renamed Table 3

---

## Author Response (AR2)

*Below are the comments from the reviewers in **black** and replies from the authors in **blue***
*Note that page and line numbers given in the replies refer to the revised version of the manuscript without tracked-changes*

**Co-Editor**

Dear authors, please find enclosed a referee report on the revised version of your manuscript, where the referee asks for a further minor revision. After correcting this minor point your paper should be ready to be accepted for publication in ACP. Additionally, I would like to ask you to consider the following suggestions for correction:

We thank the co-editor for careful reading and valuable suggestions. Below we reply point-by.point to the co-editor's suggestions.

P8, L20: The sentence in its present form is not correct. I would suggest to rephrase it as follows: "However, with our measurements sites not being tropical in a geographical sense, we refer........" or "However, since our measurements sites are not tropical in a geographical sense, we refer....":

Done (page 8 lines 21-22).

P25, Figure 2 caption: If you use for the vertical resolution of the model data the term L137, I would suggest to also use the corresponding term for the horizontal resolution, Txx.

Done (pages 25-26). Caption of Figure 3 was also modified accordingly (page 27).

P32, Figure 8 caption: The term "free" is only used for the troposphere, but not for the stratosphere. Thus, I would suggest to just write "stratosphere".

We thank the co-editor for this suggestion. We are aware that the term "free" is usually used for the troposphere but not for the stratosphere. Neverheless, we believe that the use of "free strato-sphere" is appropriate for our work. This makes clear the distinction between the confined lower stratopshere (CLS), in which air masses are confined in the Asian sector of the lower stratosphere due to the presence of the anticyclone, and the free stratopshere above the anticyclone, in which the air masses are "free" in the sense that they are not confined to the Asian sector, but rather can move around all longitudes of the global stratosphere. Therefore, we prefer to maintain the current formulation and we refrain from doing any changes.

**Anonymous Referee #2**

The authors meticulously addressed the points raised by the review(s), improving presentation as well as contents. Now the paper is easier to read and the discussion is more balanced with respect to different concepts and competing processes. Remaining uncertainties are pointed out clearly. There is one minor point. The definition of TOC contains some arbitrary elements: 50 % threshold, 2 weeks trajectory length, choice of ASMA region (section 5.2). This is ok. Some definition of confinement is needed to compare different altitudes in one dataset, or different datasets with each other. However, there seems to be no particular physical distinction of your TOC level. Why not choosing 49.5% or 15 days trajectory length? This is in contrast to LRM and CPT. One example (P13L18): The degree of confinement and its variation with altitude are certainly important for

the composition of the ASMA and possibly also for its thermodynamics, but your TOC is not a thermodynamically significant level like LRM or CPT. I suggest pointing out this difference more clearly throughout the paper.

We thank Anonymous Referee #2 for careful reading and valuable suggestion. We agree that, given the arbitrary elements in its definition, TOC is not a thermodynamically significant level like LRM and TOC. We now point out this difference at page 14 lines 27-30. Page 9 line 24 and page 13 lines 18-19 were also modified accordingly.

[revised manuscript text omitted]